# Stratospheric impacts on dust transport and air pollution in West Africa and the Eastern Mediterranean

Ying Dai [1], Peter Hitchcock [1] ✉, Natalie M. Mahowald [1], Daniela I. V. Domeisen [2,3], Douglas S. Hamilton [1,9], Longlei Li[1], Beatrice Marticorena[4], Maria Kanakidou [5,6,7], Nikolaos Mihalopoulos[5,8] & Adwoa Aboagye-Okyere[1]

Saharan dust intrusions strongly impact Atlantic and Mediterranean coastal regions. Today, most operational dust forecasts extend only 2–5 days. Here we show that on timescales of weeks to months, North African dust emission and transport are impacted by sudden stratospheric warmings (SSWs), which establish a negative North Atlantic Oscillation-like surface signal. Chemical transport models show a large-scale dipolar dust response to SSWs, with the burden in the Eastern Mediterranean enhanced up to 30% and a corresponding reduction in West Africa. Observations of inhalable particulate ($PM_{10}$) concentrations and aerosol optical depth confirm this dipole. On average, a single SSW causes 680–2460 additional premature deaths in the Eastern Mediterranean and prevents 1180–2040 premature deaths in West Africa from exposure to dust-source fine particulate ($PM_{2.5}$). Currently, SSWs are predictable 1–2 weeks in advance. Altogether, the stratosphere represents an important source of subseasonal predictability for air quality over West Africa and the Eastern Mediterranean.

The world's largest dust source is North Africa[1], where strong winds can lift large amounts of dust from bare, dry soils into the atmosphere. Once entrained into the atmosphere, North African dust is transported towards the Atlantic Ocean and/or towards the Mediterranean Sea[2,3]. Dust aerosols therefore can have vast health, visibility, environmental, and economic impacts on large population centers and industrial areas concentrated along the coasts of the Atlantic and the Mediterranean[4–7]. During episodes of Saharan dust storms that affect Europe, there is substantial evidence of the windblown desert dust's association with mortality and morbidity[8–11]. Apart from devastating

health impacts, dust also impacts the environment, transport, and infrastructure[12–14]. Monetizing these impacts can translate to hundreds of million dollars just from a single dust storm[13]. According to the UN[15], in the Middle East and North Africa (MENA) region, about 13 billion USD in Gross Domestic Product (GDP) are lost every year due to dust storms alone. According to the World Bank[13], dust pollution and dust storms cost MENA over 150 billion USD annually and over 2.5% of GDP for most countries in the region. Forecasts of hazardous air pollution caused by dust aerosols are crucial to help reduce personal exposure to outdoor air pollution, and would help decision makers to restrict

[1]Department of Earth and Atmospheric Sciences, Cornell University, Ithaca, NY 14853, USA. [2]University of Lausanne, Lausanne, Switzerland. [3]ETH Zurich, Zurich, Switzerland. [4]Laboratoire Interuniversitaire des Systèmes Atmosphériques, Universités Paris Est-Paris Diderot-Paris 7, UMR CNRS 7583, Créteil, France. [5]Environmental Chemical Processes Laboratory (ECPL), Department of Chemistry, University of Crete, Heraklion, Greece. [6]Center of Studies of Air quality and Climate Change, Institute for Chemical Engineering Sciences, Foundation for Research and Technology Hellas, Patras, Greece. [7]Excellence Chair, Institute of Environmental Physics, University of Bremen, Bremen, Germany. [8]Institute for Environmental Research and Sustainable Development, National Observatory of Athens, Pendeli, Greece. [9]Present address: Department of Marine, Earth, and Atmospheric Science, NC State University, Raleigh, NC, USA. ✉e-mail: aph28@cornell.edu

anthropogenic emissions of domestic, traffic, and industrial pollutants during predicted periods of high pollution[16]. However, across much of Africa, dust still arrives without accurate intensity warnings[17]. In many parts of Europe, today's dust models are only capable of providing operational dust forecasts for several days in advance[18,19]. Case studies of North African dust storms with strong impact over southern Europe found a common large-scale upper-level precursor (a double Rossby wave breaking process) developing 5–10 days prior to dust storm formation. Using this precursor signal as an indicator might provide early warnings of high-impact North African dust outbreaks over southern Europe at lead times of 5–10 days[20,21]. In general, the predictive lead time is limited to several days. Extending this lead time to weeks and even months would provide medical, agricultural, and nautical stakeholders more time to prepare for risks associated with airborne dust[17].

One potential approach towards filling this capability-need gap is to consider the impact of the stratosphere. Stratospheric variability is well-recognized as an important source of predictability for surface weather and climate on subseasonal to seasonal (S2S) timescales, corresponding to timescales between 2 weeks and 2 months[22–24]. A particularly extreme phenomenon of the winter stratosphere that contributes to S2S predictability at the surface is the so-called sudden stratospheric warming (SSW). An SSW corresponds to a breakdown of the stratospheric polar vortex, in which the polar stratosphere warms rapidly and the westerly circumpolar winds weaken dramatically and reverse to easterlies[25]. Currently, individual SSW events can be predicted about 1–2 weeks in advance[26] and can be detected early on with satellite observations[27]. Subsequently, SSWs tend to influence surface weather for up to a few weeks[28]. This influence allows for timescales of weeks to months to study the surface weather impacts of stratospheric forcing, pointing to their predictive value for surface weather on S2S timescales[29].

The tropospheric circulation response to SSW events is commonly characterized as an equatorward shift of the North Atlantic eddy-driven jet, corresponding to a negative phase of the North Atlantic Oscillation (NAO)[30–33]. The negative phase of the NAO is associated with precipitation extremes over Southern Europe[34] and cold air outbreaks over Northern Eurasia and the eastern United States[35–37]. Beyond its impact on surface meteorological fields such as temperature and precipitation, the NAO exerts a large-scale control on the dust transport out of North Africa[2], which makes up almost half of the dust source on Earth[38]. Despite the reported link between the NAO and African dust export, the connection between SSWs and desert dust export has not been explored, and could contribute to long-term predictions of African dust export.

In this article, we present evidence of the impact of SSWs on African dust export and the consequent impact on air quality over S2S timescales, with chemical transport models and station observations finding up to 30% changes in aerosol concentrations over West Africa and the Eastern Mediterranean caused by SSWs. In particular, SSWs induce meteorological conditions that increase northward African dust transport to the Eastern Mediterranean, while at the same time reduce African dust emission and decrease westward dust transport to West Africa. Consequently, the former leads to enhanced dust burden in the Eastern Mediterranean and the latter results in a reduction in dust burden within West Africa. These changes caused by SSWs worsen air pollution in Southern European countries such as Greece, while they help reduce air pollution in West African countries such as Senegal.

## Results

### Detecting a large-scale dust response to sudden stratospheric warming events

As described in the introduction, SSWs can lead to a tropospheric circulation response that resembles the negative phase of the NAO. In terms of the sea level pressure (SLP) field, a negative NAO corresponds to an anticyclonic anomaly at Stykkisholmur, Iceland and a cyclonic anomaly at Lisbon, Portugal[39]. Composite averages over SSW episodes (defined as the 30-day periods after the onset of SSW events; see Methods) indeed reveal a cyclonic SLP anomaly over southern Europe/northern Africa (Fig. 1a, black contours and shading), similar to that associated with a negative NAO (Fig. 1a, yellow contours). On the one hand, this cyclonic SLP anomaly corresponds to a decrease in the strength of the subtropical ridge in northern Africa and the strong pressure gradient associated with it (Fig. 1b). In this region, a weakening of the south-north pressure gradient decreases the speed of northeasterly trade winds. On the other hand, the cyclonic SLP anomaly causes strong southwesterly winds from the Sahara towards the Eastern Mediterranean (Fig. 1a, black arrows). Note that the downward influence of SSWs is more intense and far reaching than its projection on to the NAO would imply. In particular, the cyclonic SLP anomaly caused by SSWs (Fig. 1a, black contours and shading) is more strongly negative and extended further eastward than the component that can be associated with the NAO (Fig. 1a, yellow contours). This NAO-SSW comparison suggests that the downward influence of SSWs is not only longer-lasting[40] but also stronger and broader than a negative NAO. Understanding their difference—potentially due to additional remote influences that affect both the stratosphere and surface weather—warrants further investigation, but is outside the scope of the present work. Given the reported link between the NAO and desert dust export[2], the downward influence of SSWs, which is closely linked to a negative NAO (Fig. 1a), points to a potentially important role of SSWs in controlling the North African dust export.

To detect if SSWs exert a large-scale control on North African dust export, two chemical transport models with specified meteorology are used (see Methods). Owing to their spatio-temporal continuity, the simulated surface dust concentrations enable us to obtain the large-scale dust response to SSWs. The surface dust response in CESM2 features a meridionally dipolar pattern with a positive response over the Eastern Mediterranean and a negative response over wide areas of West Africa (Fig. 1c). The east-westward aligned negative response over West Africa matches geographically well with a regional belt where surface dust concentrations are among the highest during the extended winter season from November to March (Fig. 1d). The MERRA2-surface dust concentrations show a very similar dipolar response (Supplementary Fig. 2a), demonstrating the robustness of the large-scale surface dust response to SSWs.

To further quantify the dust response to SSWs, SSW-caused changes in dust pollution are obtained by comparing the dust pollution during SSW episodes with those during non-SSW episodes. For this purpose, 1000 sets of non-SSW episodes are identified (see Methods). The dust pollution is measured by the level of surface dust concentrations and the number of high dust pollution days. These days correspond to those on which the specified daily thresholds 250, 500, and 1000 $\mu g\, m^{-3}$ are exceeded (Supplementary Table 1). In the Eastern Mediterranean, SSWs lead to a 20–30% increase in the level of surface dust concentrations and a 20–40% increase in the number of high dust pollution days (Supplementary Table 1). In West Africa, decreases are simulated. The level of surface dust concentrations is reduced by 10–20%, and the number of high dust pollution days decreases between 10 and 30%, depending on the model and threshold used (Supplementary Table 1). All these changes are relative to the non-SSW mean, which is defined as an average over 1000 sets of non-SSW episodes (see Methods). The quantitative analysis shows that dust pollution at the surface is influenced to a major extent by SSWs.

The dust represents one of the main natural contributions to atmospheric particulate matter (PM), which has direct effects on people's health when inhaled. We next present results of how SSWs, via influencing dust pollution at the surface, impact mortality from exposure to dust-source fine particulate matter less than 2.5 μm in

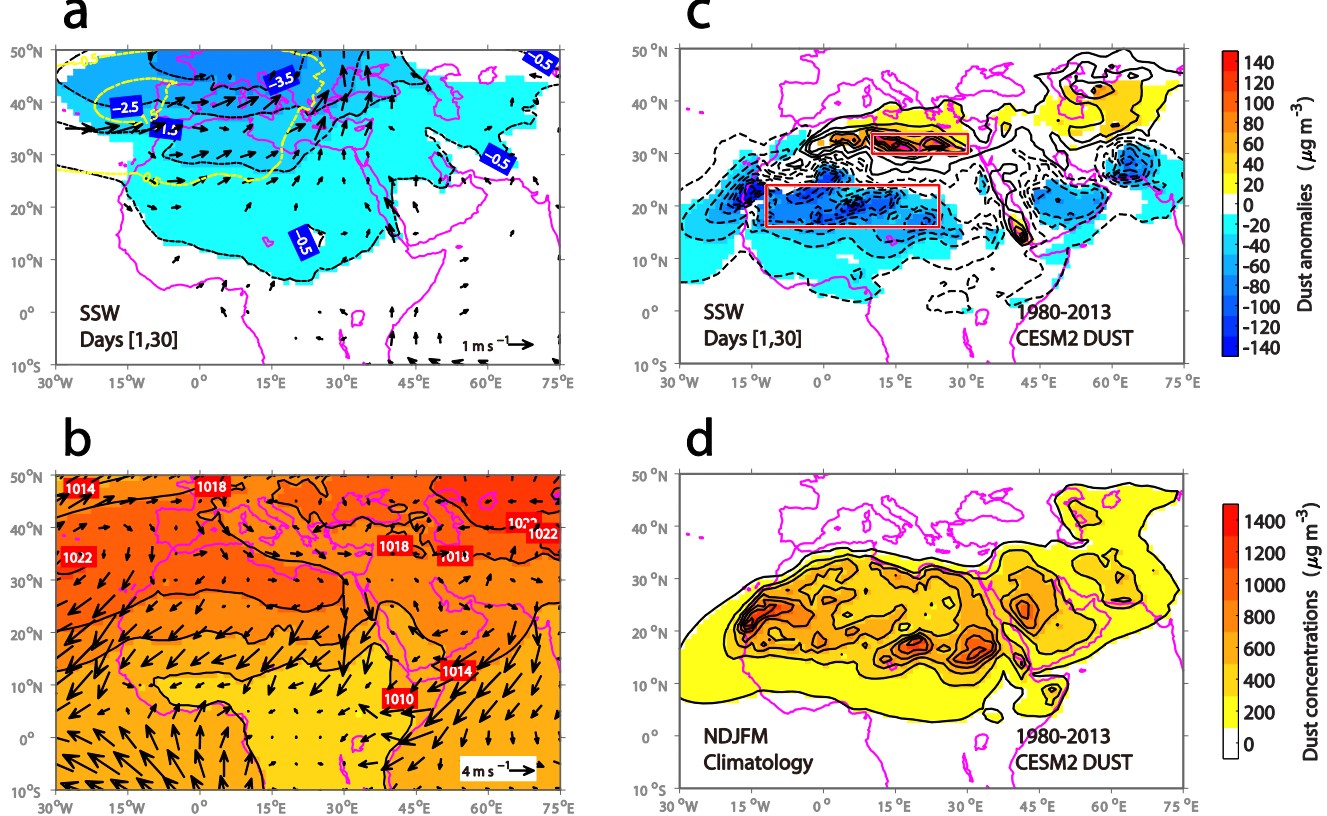

**Fig. 1 | Large-scale meteorological and dust response to sudden stratospheric warming events (SSWs). a** Composites of anomalous sea level pressure (SLP) (contours) and 10 m wind (arrows) during SSW episodes over the 1980/81-2013/14 extended winters (from November to March, NDJFM). The SLP and 10 m wind fields come from the MERRA2 reanalysis. Shadings indicate SLP anomalies that are statistically significant at the $p < 0.10$ level based on a two-tailed Monte Carlo test. Wind vectors shown are statistically significant at least for one component. Scaling for the wind vectors is given in the bottom-right corner (units: ms$^{-1}$). Yellow contours indicate the regressed SLP anomalies onto the extended wintertime (NDJFM) North Atlantic Oscillation (NAO) index (see Methods). To make them comparable to the composite SLP anomalies following SSWs, the regressed SLP anomalies are scaled by the average value of the NAO index during SSW episodes, which is slightly greater than −0.3 (a histogram is shown in Supplementary Fig. 1). **b** The corresponding extended wintertime (NDJFM) climatologies of SLP and 10 m wind fields. **c, d** The same as **a, b** but for CESM2 surface dust concentrations (units: μg m$^{-3}$). The two red boxes in panel **c** indicate the Eastern Mediterranean (30°–33.75°N, 10°–30°E) and West Africa (16.25°–23.75°N, 12.5°–23.75°E) regions with large and significant dust response to SSWs. Made with Natural Earth. Free vector and raster map data @ naturalearthdata.com.

diameter (PM$_{2.5}$) (see Methods). Generally, SSWs lead to additional premature deaths in the Eastern Mediterranean via enhancing PM$_{2.5}$ concentrations and reduced premature deaths in West Africa by decreasing PM$_{2.5}$ concentrations (Supplementary Fig. 3). On average, a single SSW can cause 2460 (±930, 1σ) premature deaths in the Eastern Mediterranean (the red box in Supplementary Fig. 3a) and prevent 2040 (±660, 1σ) premature deaths in West Africa (the blue box in Supplementary Fig. 3a), as estimated from the CESM2 output of dust-source PM$_{2.5}$. MERRA2 suggests a smaller but comparable signal, that is, a single SSW can lead to 680 (±210, 1σ) additional premature deaths in the Eastern Mediterranean and 1180 (±750, 1σ) reduced premature deaths in West Africa. The difference in the magnitude of the response in different models is consistent with the uncertainty in the dust cycle across different dust models[41]. The impact of SSWs on dust pollution mortality suggests that skillful forecasts of dust pollution during SSWs may alleviate pressure on health and social care systems.

### Meteorological causes of the dipolar dust response

Understanding the meteorological causes of the dipolar dust response is important because it provides theoretical support for the model results. As we show next, during SSW episodes, the enhanced dust pollution over the Eastern Mediterranean is linked to increased northward African dust transport, while the reduction in dust pollution within West Africa arises from a combination of suppressed African dust emission and decreased westward dust transport.

In the Eastern Mediterranean, the southwesterly winds, which facilitate northward Sahara desert dust transport, create a potential for high dust concentrations (Fig. 2a). For example, the maximum particulate matter less than 10 μm in diameter (PM$_{10}$) concentrations level for 2015 recorded at Finokalia (Crete, southern Greece) was related to a desert dust outbreak event caused by a strong southwesterly airflow[42]. During SSW episodes, for the Eastern Mediterranean, a significant increase in the fraction of time and area that the winds are southwesterly is seen (Fig. 2b), consistent with the fact that the SSW-induced cyclonic SLP anomaly is associated with southwesterly winds from the Sahara towards the Eastern Mediterranean (Fig. 1a, black arrows). In this sense, the enhanced dust burden over the Eastern Mediterranean during SSW episodes (Fig. 1c) is likely related to the increased northward Saharan dust transport induced by an increase in the fraction of southwesterly winds (Fig. 2b). In terms of the surface wind speeds, however, there are no significant changes due to the SSW (Supplementary Fig. 4a). Note that the Eastern Mediterranean is also exposed to dust originating from western Asia (mainly the Arabian Peninsula). Therefore, Arabian Desert dust imported under an easterly flow can also lead to dust intrusions into the Eastern Mediterranean[43–45]. However, during SSW episodes, a significant decrease in the fraction of easterly wind is

seen (Fig. 2b), indicating that the enhanced dust burden over the Eastern Mediterranean during SSW episodes is unlikely contributed by dust from the Arabian Desert. There is no evidence for changes in dust burden due to the SSW in the Western Mediterranean basin (Fig. 1c).

In West Africa, a decrease in surface wind velocity is associated with a lower probability of high dust concentrations (Fig. 2c), as the slower winds pick up and transport less dust from the Sahara to West Africa[46]. In particular, extremely intense $PM_{10}$ concentrations at Cinzana (Mali) are recorded only for higher than $5\,ms^{-1}$ local wind velocities[47]. During SSW episodes, for West Africa, a weakening of surface wind velocities is seen (Fig. 2d), as can be inferred from the

decreased probability of the high wind velocities (higher than $5\,ms^{-1}$) and the increased probability of the low wind velocities (lower than $4\,ms^{-1}$) (Fig. 2d). In this case, the reduced dust burden within West Africa during SSW episodes (Fig. 1c) may be linked to the suppressed dust emission and reduced westward dust transport in response to a reduction in surface wind velocities (Fig. 2d). In terms of the surface wind directions, however, there are no significant changes due to the SSW (Supplementary Fig. 4b).

To sum up, during the extended boreal winter season (NDJFM) (Fig. 3a), the subtropical ridge tends to be located over northern Africa (Fig. 3a, red contour line; Supplementary Fig. 5a, thick contour line), and is associated with strong northeasterly trade winds (Fig. 3a, black

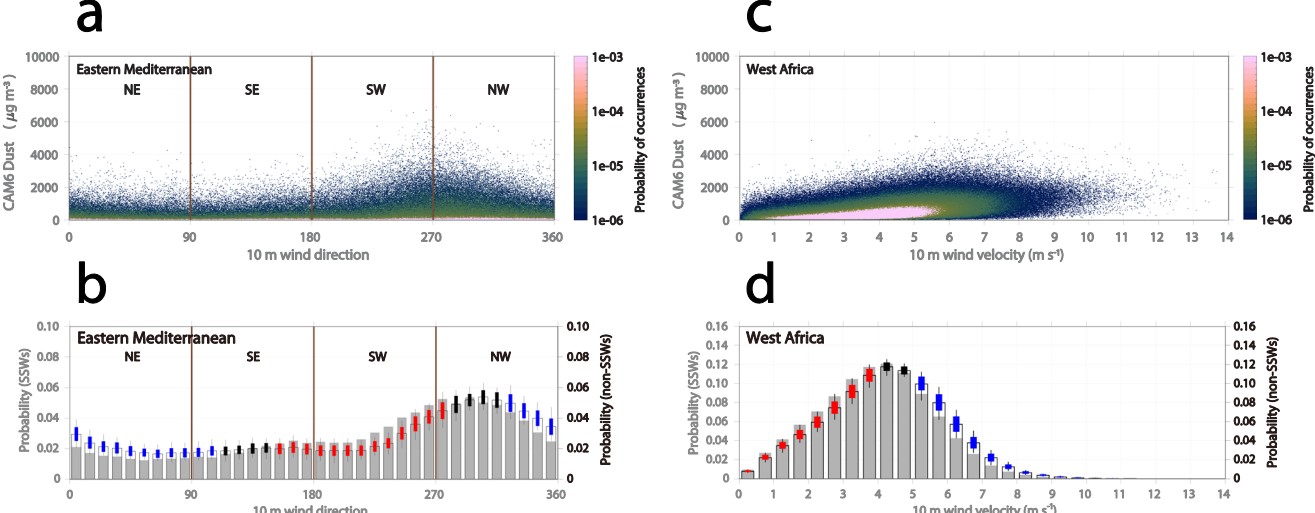

**Fig. 2 | Observed changes in surface wind fields within the Eastern Mediterranean and West Africa.** The Eastern Mediterranean and West Africa correspond to the two red boxes shown in Fig. 1c. **a** Bivariate histogram of surface dust concentrations (from the CESM2 simulation) and surface wind directions (from the MERRA2 reanalysis) at all grid points within the Eastern Mediterranean during 1980/81-2013/14 extended winters (from November to March, NDJFM), with colors indicating the probability of occurrences falls within each bin. For surface wind directions, "NE", "SE", "SW", and "NW" indicate north-easterlies, south-easterlies, south-westerlies, and north-westerlies, respectively. **b** Histogram of surface wind directions at all grid points within the Eastern Mediterranean during SSW episodes (gray filled bars) and during 1000 sets of non-SSW episodes (see Methods). For

1000 sets of non-SSW episodes, black unfilled bars show the mean and black whiskers show the spread. The boxes (filled in black, red, and blue colors) indicate the 5th and 95th percentiles. The boxes are filled in red and blue when SSW-caused increases and decreases are statistically significant at the $p < 0.10$ level based on a two-tailed Monte Carlo test, respectively. **c, d** The same as **a**, **b** but for surface wind velocities (from the MERRA2 reanalysis) at all grid points within West Africa. To generate the bivariate histogram plots, we partition the surface dust concentrations into 500 bins of width $20\,\mu g\,m^{-3}$ within [0, 10,000], the surface wind direction into 720 bins of width $0.5°$ within [0, 360], and the surface wind velocity into 700 bins of width $0.02\,ms^{-1}$ within [0, 14], respectively.

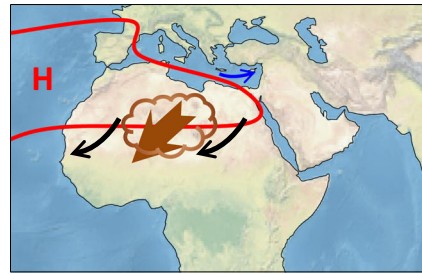

a **Wintertime Climatology**

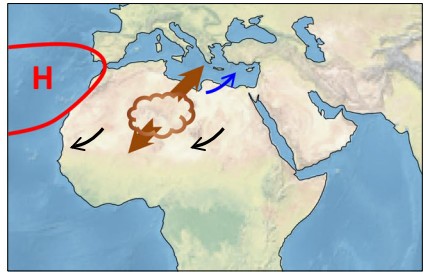

b **Days [1,30] after SSWs**

**Fig. 3 | Schematic presentation of the stratospheric impacts on tropospheric circulation and North African dust. a** The extended wintertime (from November to March, NDJFM) climatology of the tropospheric circulation and North African dust. **b** The same as panel a but for the sudden stratospheric warming (SSW) episodes. Various quantities used to illustrate the dust response include the subtropical ridge (red contour line; see Supplementary Fig. 5 for a description), surface winds (black and blue arrows), dust emission (brown cloud-shaped pattern), and dust transport (brown arrows). As can be seen, during the extended winter season (NDJFM) (panel a), the subtropical ridge tends to be located over northern Africa.

The associated northeasterly trade winds pick up and transport much of the dust aerosols southwestward. Meanwhile, the Eastern Mediterranean is under the influence of midlatitude westerlies. During SSW episodes (panel **b**), the subtropical ridge retreats to the North Atlantic Ocean. Weaker northeasterly trade winds pick up and transport less dust aerosols southwestward, while southwesterly winds are more likely to occur over the Eastern Mediterranean, increasing northward dust transport. Made with Natural Earth. Free vector and raster map data @ naturalearthdata.com.

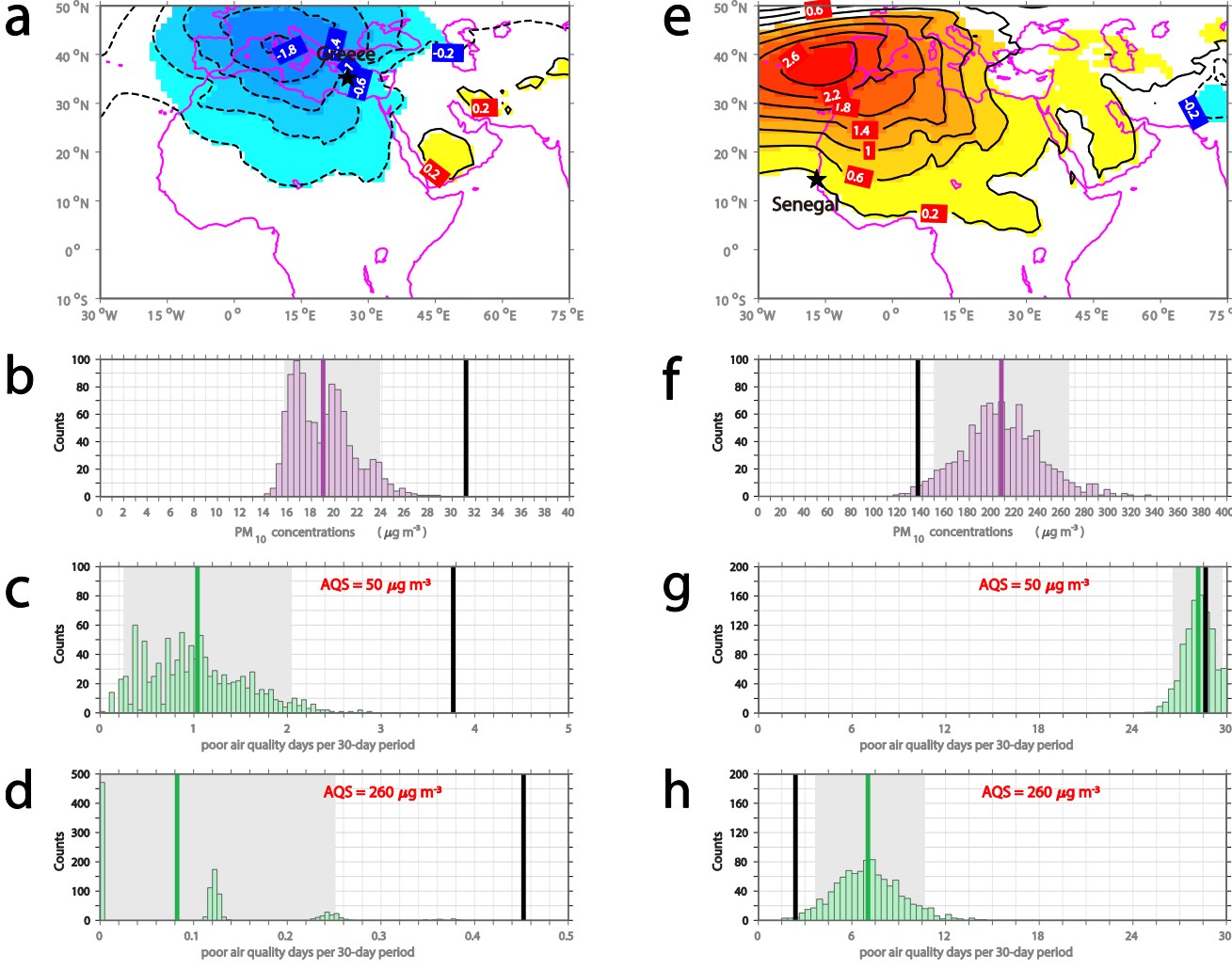

**Fig. 4 | Observed air quality response to sudden stratospheric warming events (SSWs). a** Regression of anomalous sea level pressure (SLP) (from the MERRA2 reanalysis) onto the station measurements of daily time-series of surface inhalable particulate (PM$_{10}$) concentrations from Finokalia, Greece. Warm and cold shadings indicate positive and negative anomalies that are statistically significant at the $p < 0.10$ level based on a Student's $t$-test, respectively. Histogram of **b** the level of PM$_{10}$ concentrations (units: µg m$^{-3}$) at Finokalia, Greece and **c, d** the expected number of poor air quality days per 30-day period for two air quality standards

(AQS), during 1000 sets of non-SSW episodes (colored bars, see Methods). Thick vertical colored line indicates the non-SSW mean; gray shading indicates the 5th–95th percentile confidence interval derived from the non-SSW spread. Thick vertical black line indicates the SSW composite. In panels **c, d**, poor air quality days are defined when the daily-mean PM$_{10}$ concentrations exceed the daily limit of **c** 50 µg m$^{-3}$ and **d** 260 µg m$^{-3}$, respectively. **e–h** The same as **a–d** but for station observations of surface PM$_{10}$ concentrations from M'Bour, Senegal. Made with Natural Earth. Free vector and raster map data @ naturalearthdata.com.

arrows). In arid regions such as North Africa, strong trade winds favor dust emissions to the atmosphere (Fig. 3a, brown cloud-shaped pattern) and help transport much of the dust aerosols southwestward (Fig. 3a, brown arrow). Meanwhile, the Eastern Mediterranean is under the influence of midlatitude westerlies (Fig. 3a, blue arrow). During SSW episodes, a cyclonic SLP anomaly is observed in the region of southern Europe/northern Africa, which is a typical pressure center of action of the negative NAO phase (Fig. 1a). Over the Eastern Mediterranean, the cyclonic SLP anomaly brings southwesterly winds (Fig. 3b, blue arrow), which favor northward dust transport (Fig. 3b, brown arrow, pointing northeast) and thus enhance dust burden in the Eastern Mediterranean (Fig. 1c). Over the continent to the south, in the presence of the cyclonic SLP anomaly, the subtropical ridge retreats to the North Atlantic Ocean (Fig. 3b, red contour line; Supplementary Fig. 5b, thick contour line). This corresponds to a weakening of the northeasterly trade winds (Fig. 3b, black arrows), which results in a decrease in African dust emission (Fig. 3b, brown cloud-shaped pattern) and a reduction in westward dust transport (Fig. 3b, brown arrow,

pointing southwest). As a result, reduced dust burden is seen in West Africa (Fig. 1c).

## Observational evidence of the dipolar dust response

Changes in dust transport and emission will shape changes in human environment. Of particular importance will be those changes in air quality impacting the local population of the affected regions. Here we use a range of observational datasets to quantify the impact of SSWs on the air quality of countries in Southern Europe and West Africa.

First, we use the daily-mean PM$_{10}$ concentrations collected at background stations in Greece and Senegal (see Methods). Overall, the regressed SLP anomaly onto the PM$_{10}$ time-series at the two stations suggests that, higher-than-normal PM$_{10}$ concentrations in Greece tend to occur under the influence of a cyclonic SLP anomaly (Fig. 4a), while higher-than-normal PM$_{10}$ concentrations in Senegal tend to occur in the presence of an anticyclonic SLP anomaly (Fig. 4e). Given the fact that a cyclonic SLP anomaly is seen following SSWs (Fig. 1a), above-normal PM$_{10}$ concentrations (worse air quality) in Greece, and below-

normal $PM_{10}$ concentrations (better air quality) in Senegal can be expected during SSW episodes.

To examine whether this expectation holds, two air quality indicators are considered: (a) the level of $PM_{10}$ concentrations, and (b) the expected number of poor air quality days when $PM_{10}$ concentrations exceed the given thresholds (two thresholds are considered, see Methods). In Greece, all air quality indicators during SSW episodes (black vertical lines in Fig. 4b–d) lie entirely beyond the upper end of the non-SSW spread (color histograms in Fig. 4b–d), indicating that the air quality in Greece is worsened significantly during SSW episodes. In particular, the level of $PM_{10}$ concentrations reaches $31 \mu g\, m^{-3}$ (black vertical line in Fig. 4b), an increase of 64% relative to the non-SSW mean ($19 \mu g\, m^{-3}$; purple vertical line in Fig. 4b); the expected number of poor air quality days per 30-day period exceeding $50 \mu g\, m^{-3}$ reaches 3.8 days (black vertical line in Fig. 4c), roughly four times the non-SSW mean (one day; green vertical line in Fig. 4c); the expected number of poor air quality days per 30-day period exceeding the higher threshold of $260 \mu g\, m^{-3}$ also increases by a factor of four (Fig. 4d), though $PM_{10}$ concentrations at this station rarely reach this level. In Senegal, the level of $PM_{10}$ concentrations and the expected number of poor air quality days exceeding the threshold of $260 \mu g\, m^{-3}$ decrease significantly during SSW episodes (Fig. 4f, h), indicating that the air quality in Senegal is improved considerably due to the SSW. In particular, the level of $PM_{10}$ concentrations is $136 \mu g\, m^{-3}$ (black vertical line in Fig. 4f), a decrease of 34% relative to the non-SSW mean ($207 \mu g\, m^{-3}$; purple vertical lines in Fig. 4f); the expected number of poor air quality days exceeding the threshold of $260 \mu g\, m^{-3}$ decreases to 2.4 days (black vertical line in Fig. 4h), about 34% of the non-SSW mean (7.0 days; green vertical line in Fig. 4h). The lower threshold of $50 \mu g\, m^{-3}$ is nearly always exceeded at this station, and no significant changes are seen during SSW episodes in this case (Fig. 4g). We note nonetheless that changes in the expected number of poor air quality days during SSW episodes are robust across a wide range of thresholds (Supplementary Fig. 6), and that exposure to air pollutants at any level has observable health impacts[48–50].

To sum up, SSWs lead to a significant increase in the level of $PM_{10}$ concentrations in Greece and a significant decrease in the level of $PM_{10}$ concentrations in Senegal. The oppositely signed air quality changes in response to SSWs (worsened in Greece and improved in Senegal) detected in station observations of $PM_{10}$ concentrations show good agreement with the dipolar dust response derived from the modeled datasets (Fig. 1c; Supplementary Fig. 2a).

Using the ground-based Aerosol Robotic Network (AERONET) direct measurements of aerosol optical depth (AOD) yields responses similar to the above results, which consist of above-normal AOD (increased by 10%) in Greece and below-normal AOD (reduced by 11%) in Senegal (Supplementary Table 2). The consistency of model results with station observations lends strong support to the impacts of SSWs on air pollution in Southern Europe and West Africa.

## Discussion

Saharan dust intrusions are a recurrent problem threatening public health and economic well-being of a large population in Southern Europe and West Africa. Forecasting such high impact but short duration events poses profound scientific challenges[51]. Major progress in the basic understanding and predictive capability can take decades[17]. Today, many developing countries across Africa do not have the computing power to forecast air quality[17]. In many parts of Europe, most forecasts of air quality only cover 2–5 days. Extending this to longer term forecasts would give communities more time to prepare for risks associated with air pollution.

Using a wide range of datasets, our results demonstrate the existence of large-scale stratospheric impacts on North African dust emission and transport on S2S timescales with the consequent impact on air quality, implying the potential for long-term dust and air

pollution predictions. In particular, chemical transport models consistently suggest a large-scale dipole dust response to SSWs, which consists of an enhanced dust signal in the Eastern Mediterranean and a reduction in West Africa. The dust pollution changes caused by SSWs can further impact premature deaths from exposure to $PM_{2.5}$. According to our estimate, a single SSW, on average, can cause 680–2460 premature deaths in the Eastern Mediterranean via dust-source $PM_{2.5}$ enhancement and prevent 1180–2040 premature deaths in West Africa through dust-source $PM_{2.5}$ reduction. The changes from an SSW event in dust pollution mortality are comparable to the changes in cold weather mortality in the UK caused by an SSW, which is 620 additional deaths per event[52]. Using station observations of $PM_{10}$ concentrations and AERONET AOD yields very similar air pollution responses to SSWs, including worsened air quality in Greece and improved air quality in Senegal, in line with the dipolar dust response derived from models. The stratospheric impacts on dust transport and air pollution at the surface are driven by the role of SSWs in establishing a negative NAO-like signal at the surface. This NAO phase is typically associated with a weakened subtropical ridge, which, through increasing southwesterly winds in the Eastern Mediterranean and weakening northeasterly trade winds in West Africa, creates large-scale meteorological conditions favorable for the dipolar dust response. The agreement between models and the consistency with station observations and the meteorological causes lend strong support to the stratospheric impacts on air pollution at the surface. Overall, the changes in air pollution in response to SSWs range between 10 and 30% (with a couple of outliers reaching 60%) compared to the non-SSW mean, depending on the region of concern and on the dataset.

The oppositely signed air quality changes between the Eastern Mediterranean and West Africa in response to SSWs suggest that the occurrence of SSWs is not always a tale of destruction. While SSWs act to worsen air pollution in Southern European countries such as Greece, for West African countries such as Senegal, air quality improves in response to SSWs although $PM_{10}$ concentrations remain at very high levels. In addition to weak vortex events such as SSWs, there are also strong vortex events that correspond to tropospheric weather regimes of opposite sign to those following SSWs[28]. Compared to SSWs, the strong vortex events might have the opposite effect on the African dust export and strongly worsen air quality in West African countries instead. Furthermore, so-called final warming events[53] in spring can lead to same-signed surface anomalies as SSW events[54], and may therefore extend the surface impacts of the stratosphere into spring, with similar effects on dust transport as SSW events. We note that stratospheric extreme events do not necessarily cause dust storms, rather they act over a period of weeks to months to increase the probability of meteorological conditions favorable for high dust pollution in certain regions. This knowledge helps assess the likelihood that high dust pollution may occur and thus can provide an advance warning for adverse conditions.

Currently, extreme stratospheric polar vortex events are predictable at 1–2 weeks lead times in operational S2S prediction systems[26]. Furthermore, once an extreme stratospheric event occurs, it can lead to changes at surface air pollution that can last for weeks to months. In this context, the stratosphere represents a non-negligible source of predictability for surface air pollution on S2S timescales, particularly for Southern Europe and West Africa. These findings provide a window of opportunity for advanced air quality warnings at lead times of weeks to months based on the forecast of extreme stratospheric events, and call for consideration of the stratospheric variability in air quality forecasting systems.

## Methods
### Datasets
The meteorological fields examined include daily-mean sea level pressure (SLP), 10 m zonal wind (U10 m), and 10 m meridional wind

(V10 m) from MERRA2 (the Modern-Era Retrospective Analysis for Research and Applications, version 2)[55]. These meteorological fields have a 1.25° × 1.25° latitude-longitude resolution.

For air quality related fields, we use daily data from a wide range of sources, including model outputs, station observations, and aerosol reanalysis. Firstly, we examine simulated daily-mean surface dust concentrations from CESM2 (Community Earth System Model, version 2)[56] and MERRA2 (The MERRA2 Aerosol Reanalysis)[57]. Note that the dust field from MERRA2 is not directly constrained by the aerosol assimilation included in MERRA2 although the overall optical depth is constrained[57]. The dust field from MERRA2 is thus to a large extent a model-based product, just like that from CESM2[56]. Dust in CESM2 and MERRA2 simulations is modeled via the chemical transport models DEAD (Dust Entrainment And Deposition model)[58] and GOCART (Goddard Chemistry, Aerosol, Radiation, and Transport model)[59], respectively. Both CESM2 and MERRA2 simulations are driven by MERRA2 meteorology. The CESM2 output covers the years 1980–2014 with a latitude-longitude resolution at -0.9° × 1.25°. The MERRA2 output covers the years 1980–2020 with a latitude-longitude resolution at 1.25° × 1.25°. Owing to their spatio-temporal continuity, the model outputs enable us to obtain the large-scale dust response to stratospheric extreme events.

Secondly, we present a verification of the findings derived from the model output with station observations of surface $PM_{10}$ concentrations and aerosol optical depth (AOD) (see Supplementary Fig. 7 for the station location and Supplementary Table 3 for the temporal coverage). The $PM_{10}$ concentrations are measured at four stations, including three sites in West Africa (M'Bour, Senegal; Cinzana, Mali; Banizoumbou, Niger) from INDAAF (International Network to study Deposition and Atmospheric composition in Africa) and one site at Finokalia (southern Greece) from Finokalia Atmospheric Observatory. The station AOD at the aforementioned four stations comes from AERONET (Aerosol Robotic Network)[60]. Unlike the model outputs, the observational data are not recorded everyday due to network and/or power failures, instrument maintenance, etc. In this study, the days with missing values are discarded. This is unlikely to affect the representativity of the measurements given their high recovery rate[47]. Note that the potential impacts of sampling error from short observational records and limited background stations may represent a potential source of uncertainty in observed air quality responses (Supplementary Note 1).

Thirdly, the gridded AOD from the MERRA2 aerosol reanalysis[57] is analyzed. Unlike the MERRA2 dust mentioned above, the MERRA2 AOD is directly constrained by the aerosol assimilation and therefore is a blend of observations with model hindcasts[57]. To enable evaluation against the station AOD from AERONET, the MERRA2 AOD time-series at the grid point closest to each AERONET station is evaluated.

For the gridded datasets, daily anomalies at each grid point are obtained by subtracting the seasonal cycle on that calendar day. The seasonal cycle is defined as the mean and first three Fourier harmonics of the daily climatology to remove the high-frequency sampling variability, following previous studies[61–63].

## Sudden stratospheric warmings

Sudden stratospheric warming events (SSWs) are defined by the reversal of the zonal-mean westerlies at 60°N and 10 hPa[30]. The first day on which the daily-mean zonal-mean zonal wind at 60°N and 10 hPa is easterly, is defined as the central date of the warming. Using the daily-mean zonal-mean zonal wind at 60°N and 10 hPa from the MERRA2 reanalysis, 24 SSW events are identified in the 40 extended winters (November–March, NDJFM) during 1980/81-2019/20 (Supplementary Table 4) where the MERRA2 dust is available. Of these, 22 SSWs occur during the 1980/81-2013/14 extended winters (NDJFM) when the CESM2 dust is available. The definition of SSWs is insensitive to the reanalysis dataset because the central dates for SSWs in each reanalysis

are in good agreement for the analyzed time period[64]. The impact of SSWs on the near-surface circulation is often described as resembling a negative phase of the North Atlantic Oscillation (NAO)[30–33]. In this study, the NAO index is defined from the difference between normalized SLP between Lisbon, Portugal, and Stykkisholmur, Iceland[39]. This definition has been used in a previous study to demonstrate the role of the NAO in controlling the North African dust export[2].

## SSW episodes and non-SSW episodes

To examine the tropospheric response to SSWs, we define "SSW episodes" as the 30-day periods after the onset of SSW events, because this is when the downward influence of SSWs on the troposphere can be most clearly identified[28,31]. For comparison, we define "non-SSW episodes" as the 30-day periods after the onset of non-SSW events. Each set of non-SSW events have the equivalent central dates as the observed SSWs but are randomly taken from extended winters (NDJFM) without an SSW. To account for the uncertainty in random sampling, 1000 sets of non-SSW events are generated. As reference, the mean of 1000 sets of non-SSW episodes (hereafter referred to as "the non-SSW mean") is used to detect by how much the air pollution during SSW episodes differs from that during non-SSW episodes, and the spread of the 1000 sets of non-SSW episodes (hereafter referred to as "the non-SSW spread") is used to determine to what extent the changes caused by SSWs are statistically significant.

## Premature mortality estimation

SSW events are associated with changes in surface dust concentrations, which impact air quality and human health. After the occurrence of SSWs, changes in mortality from exposure to $PM_{2.5}$ are estimated as[65]: $\Delta_{Mort} = M_b \times N_{pop} \times [\exp(\beta \Delta X) - 1]$, where $M_b$ is the baseline mortality obtained from the Global Burden of Disease (GBD) study, $N_{pop}$ is the exposed population obtained from the Center for International Earth Science Information Network, $\beta$ is the concentration-response factor for mortality due to short-term exposure to $PM_{2.5}$ and is based on a 2-day averaged anomaly in $PM_{2.5}$ concentrations (a $10 \, \mu g \, m^{-3}$ increase in 2-day averaged $PM_{2.5}$ was found to be associated with a 0.98% increase in total mortality[66]), $\Delta X$ is the 2-day averaged anomaly in $PM_{2.5}$ concentrations over days [1, 30] following SSW events (22 SSWs for CESM2 and 24 SSWs for MERRA2). Here, the model output of surface $PM_{2.5}$ concentrations specific to dust aerosols is used. In this way, the estimated mortality changes are attributable to SSWs given their impact on surface dust aerosols. There are uncertainties in the $\beta$ deduced as the concentration-response factor, as well as what baseline mortality to use in the exposure by gender, age or comorbidities, but here we use a standard relationship and standard mortality values used for air pollution impacts[66].

Mortality changes are calculated at each grid point for each of the 15, 2-day periods over days [1, 30] following the occurrence of SSWs. The 15 here indicates the number of 2-day periods over days [1, 30] after the onset of SSWs. For example, the 1st 2-day period corresponds to days [1, 2], the 2nd 2-day period corresponds to days [3, 4], the 3rd 2-day period corresponds to days [5, 6], and the 15th 2-day period corresponds to days [29, 30]. For each SSW event, the mortality changes at each of the 15, 2-day periods are added together to obtain an estimate of the excess deaths that occurred during the SSW event. The excess deaths for each SSW event are further used to calculate the average excess deaths per event (Supplementary Fig. 3a, c). Similar to the dipolar dust-source $PM_{2.5}$ response pattern (Supplementary Fig. 3b, d), the mortality changes also exhibit a dipolar pattern but with the centers moving towards large population centers (Supplementary Fig. 3a, c). The mortality changes estimated from CESM2 (Supplementary Fig. 3a) and MERRA2 (Supplementary Fig. 3c) are generally comparable although CESM2 suggests larger changes than MERRA2. In fact, across the models participating in the sixth phase of the Coupled Model Intercomparison Project (CMIP6), there is a wide range in dust

emissions wherein CESM2 is at the higher end while MERRA2 is at the lower end[41]. In this sense, our results obtained from CESM2 and MERRA2 represent the diversity in dust emission and therefore provide a robust evaluation of the possible health impacts of SSWs.

For each of the designated regions (the red and blue boxes in Supplementary Fig. 3a, c), the mortality changes at each grid point within the region and each of the 15, 2-day periods for each SSW event are added together to get an estimate of the total excess deaths that occurred during each SSW event. This gives 22 individual numbers for CESM2, one per SSW event (or 24 individual numbers for MERRA2). These 22 numbers obtained from CESM2 (or 24 numbers from MERRA2) are further used to calculate the average excess deaths per event, as well as a standard deviation across events.

### Air quality indicators

While the smaller particles ($PM_{2.5}$) have been used in the previous section, because observations of these are not available we focus on observational records of $PM_{10}$ in this section as an indicator of air quality because long-term observations of $PM_{10}$ are available for the studied areas. Using the daily-mean surface $PM_{10}$ concentrations, two air quality indicators are defined: (a) the level of $PM_{10}$ concentrations, and (b) the expected number of poor air quality days per 30-day period. Poor air quality days are defined when the daily-mean $PM_{10}$ concentrations exceed the given air quality standard. In this study, for each station, two air quality standards are used. These include the European Union air quality standard (daily-mean $PM_{10}$ concentrations of 50 µg m$^{-3}$) and a higher threshold (daily-mean $PM_{10}$ concentrations of 260 µg m$^{-3}$). The latter is the National Air Quality Standard for Senegal[67]; it has also been used in a previous study as the air quality standard for Tunisia, which is located close to desert dust source[68]. Note that exposure to permissible concentrations of air pollutants still has observable health impacts[48–50], indicating that the air quality standards should be regularly reviewed and revised as new scientific evidence emerges on adverse effects on public health and the environment.

Here is an example of how we compute the two air quality indicators. At Finokalia (Greece), the daily-mean $PM_{10}$ concentrations are collected between years 2004–2019, wherein 9 SSW events took place (Supplementary Table 3). Regarding the first air quality indicator, the level of $PM_{10}$ concentrations during SSW episodes is computed as the average of daily-mean $PM_{10}$ concentrations over the 270 days following the 9 SSW events (30 days per SSW event). The level of $PM_{10}$ concentrations during each set of non-SSW episodes is computed in the same way. To obtain the second air quality indicator, we first count the number of poor air quality days during SSW episodes and during each set of non-SSW episodes. However, a direct comparison of the number of poor air quality days between SSW and non-SSW episodes would not be fair if the number of days with available data differs between SSW and non-SSW episodes, given the fact that the observational data are not recorded everyday due to network and/or power failures, instruments maintenance, etc. To account for this uncertainty, the number of poor air quality days is divided by the number of available data days to obtain the rate of poor air quality days. To facilitate understanding, the rate of poor air quality days is multiplied by 30 to obtain the expected number of poor air quality days per 30-day period. Note that the expected number of poor air quality days obtained in this way is not necessarily integer.

To assess whether SSWs are able to cause changes in the air quality, the air quality indicators during SSW episodes are compared to the distribution of those during non-SSW episodes. The differences, if any, are expected to arise from the downward influence of SSWs.

### Statistical significance assessment

Statistical significance of the regression analysis used in this study is evaluated using a two-tailed Student's $t$-test, where the effective degrees of freedom are calculated following equation (1) of ref. 69.

Statistical significance of the composite analysis based on SSWs is assessed with a two-tailed Monte Carlo test. For anomalous fields (the seasonal cycle is subtracted), one thousand sets of composites are computed around $N$ randomly chosen 'central dates' from the pool of days during all extended winters (NDJFM) with available data, where $N$ is equal to the number of SSWs involved in the composite analysis. For full fields (the seasonal cycle is not subtracted), one thousand sets of composites based on non-SSW events are computed, so as to account for the influence of seasonal cycle in the full fields. The composite fields based on SSWs are considered significant (at the $p < 0.10$ level by a two-sided test) if it lies outside of the 5th–95th percentile range derived from the randomly generated composites.

### Data availability

The MERRA2 data used in this study come from https://gmao.gsfc.nasa.gov/reanalysis/MERRA-2/. The CESM2 data are available from the Cornell eCommons repository (https://doi.org/10.7298/xqqj-qk90). The $PM_{10}$ concentrations measured at three stations in West Africa (M'Bour, Senegal; Cinzana, Mali; Banizoumbou, Niger) come from INDAAF. The $PM_{10}$ concentrations measured at Finokalia (southern Greece) come from Finokalia Atmospheric Observatory (https://finokalia.chemistry.uoc.gr/). The station AOD at the aforementioned four stations comes from AERONET. The baseline mortality used in this study is obtained from the Global Burden of Disease (GBD) study (http://ghdx.healthdata.org/gbd-results-tool), the exposed population is obtained from the Center for International Earth Science Information Network (https://sedac.ciesin.columbia.edu/data/set/gpw-v4-population-count-rev11).

### Code availability

The CESM2 model is freely available to the scientific community. The code can be accessed through a public GitHub repository (https://github.com/ESCOMP/CESM).

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

## Acknowledgements

Y.D. acknowledges support from Cornell University. Support from the Swiss National Science Foundation through projects PP00P2_170523 and PP00P2_198896 to D.I.V.D. is gratefully acknowledged. M.K. and N.M. acknowledge support through the project PANACEA (MIS 5021516) funded by the Operational Programme "Competitiveness, Entrepreneurship and Innovation" (NSRF 2014-2020) and co-financed by Greece and the European Union Regional Development Fund. The French National Observatory Service INDAAF is supported by the INSU/CNRS, the IRD (Institut de Recherche pour le Développement), and the Observatoires des Sciences de l'Univers EFLUVE and Observatoire Midi-Pyrénées. We would like to thank the PIs and operators from France, Niger, Mali, and Senegal for maintaining the INDAAF stations (https://indaaf.obs-mip.fr).

## Author contributions

P.H. and D.I.V.D. initially conceived of the study. Y.D., P.H., N.M.M., and D.S.H. jointly designed the study. Y.D. performed the data analysis, created the figures, and wrote the manuscript. D.S.H. provided the CESM2 simulation and suggested the mortality calculation. L.L. suggested the examination of MERRA2 aerosol reanalysis. B.M., M.K., and N.M. provided the observational $PM_{10}$ concentrations. A.A.-O. calculated the mortality changes. All authors discussed the results and edited the manuscript.

## Competing interests

The authors declare no competing interests.
