## [Peer Review File · Nature Communications]

Stratospheric impacts on dust transport and air pollution in West Africa and the Eastern MediterraneanReviewer #1 (Remarks to the Author):

-What are the noteworthy results?

The sudden stratospheric warming episodes set up a favorable environment at the tropospheric lower level over the North African region that enhances (suppresses) northward (southwestward) transport of dust in the Mediterranean region (West Africa). Subsequently, the transported dust leads to increased (decreased) premature death in the Mediterranean region (West Africa). The current work provides a way that can predict the impact of extreme stratospheric warming events on dust-related air pollution over southern Europe and West Africa very early (weeks to months before).

- Will the work be of significance to the field and related fields? How does it compare to the established literature? If the work is not original, please provide relevant references.

Yes. The current dust emission and transport modeling framework cannot accurately predict dust transport beyond a few days. So finding/understanding the way to predict dust transport out of the source region (here North Africa), as a result of sudden extreme stratospheric phenomena, well before (~weeks to months) is highly significant work in this field.

- Does the work support the conclusions and claims, or is additional evidence needed? Yes, conclusions and claims are supported by the presented results/analyses.

- Are there any flaws in the data analysis, interpretation and conclusions? - Do these prohibit the publication or require revision?

I do not find any significant flaws in the data analysis that prohibit the publication. However, I have pointed out a few minor comments that need to be addressed before publication. These minor comments are provided below.

- Is the methodology sound? Does the work meet the expected standards in your field? Yes.

- Is there enough detail provided in the methods for the work to be reproduced?

Yes.

Formal Review:

This article presented how sudden stratospheric warming (SSW) events enhance dust burden in the Mediterranean region and suppress dust burden in the West African region. Additionally, the article presented the importance of SSW signals for early prediction (~weeks to months before) of dust-related air pollution conditions in the Mediterranean and the West African regions. These forecasting strategies/signals play an important role as the current operational dust forecasting models cannot accurately forecast dust transport beyond a few days' limit. However, recent work has also been done in understanding the large-scale upper-level precursors (Rossby Wave Breaking signals) critical for organizing large-scale North African dust storms and subsequent long-range transport of dust to southern Europe. So, I encourage authors to look at these latest developments/findings that can provide an early signal of a high-impact dust outbreak.

I find this article very interesting and most of the interpretations are plausible. The paper is well written and the results and analyses are properly organized. However,

there are a few minor issues that need to be addressed. Overall, I recommend this article for publication after addressing the minor comments shown below.

Minor comments

1. Line 23-24

"Apart from devastating health impacts, dust also impacts the environment, transport, and infrastructure".

Please provide reference(s) to support the impact of dust on the environment, transport, and infrastructure.

2. Line 30-31

"In many parts of Europe, today's dust models are only capable of providing operational dust forecasts for several days in advance^{16,17}. Extending this to weeks"

Here it is worthy to note the latest developments and findings, the large-scale upper-level precursors (especially Rossby Wave Breaking signals), that can be a good indicator for early prediction of high-impact North African dust outbreaks over southern Europe.

3. Line 34

"consider the impact of the stratosphere"

Is the impact of the stratosphere or the impact of the stratospheric phenomena. Please clearly state.

4. Line 51-52

"30% changes in aerosol concentrations caused by SSWs."

Where aerosol concentration is changing? Please clarify this line.

5. Line 52-53

"SSWs induce those meteorological conditions that increase northward dust transport to the Mediterranean, while at the same time they inhibit African dust emission and decrease westward dust transport to West Africa."

Is northward transport of dust solely African dust? I believe yes. If this is true, please use "northward African dust transport" just to make it consistent with your claim of inhibition of African dust emission later on.

5. Line 87-90

"In the Mediterranean, SSWs lead to a 15-30% increase in the level of surface dust concentrations and a 20-40% increase in the number of high dust pollution days (Supplementary Table 1). In West Africa, decreases are simulated. The level of surface dust concentrations is reduced by 10-20%, and the number of high dust pollution days decreases between 10-30%, depending on the model and threshold used (Supplementary Table 1)."

As these % change values are exact without uncertainties (Supplementary Table S1). So I suggest using exact value ranges. For example, SSWs lead to an increase in dust concentration by 19-28% instead of 15-30%.

6. Line 128

"during boreal winter (Fig. 3a)"

Is this boreal winter or extended boreal winter months (NDJFM) that you have considered here? If not, please specify boreal winter months (which I believe comprise different months compared to "NDJFM"). In Figure 3a, you have mentioned: "(a) Wintertime climatology of tropospheric circulation and North African dust." Is this wintertime consist "NDJFM" months or the boreal winter months? This need to be clearly mentioned because boreal winter months and extended boreal winter months are different here.

7. Line 239

"daily mean zonal mean zonal wind".

It should be "daily mean zonal wind"?

8. Line 282

"While in general smaller particles (PM2.5) are more important for health effects".

Please provide reference(s) here.

9. Line 399

"DUST STORMS URL"

Possibly full stop is missing here.

10. Line 463-464

"search: Atmospheres 114 (2009). URL

<https://agupubs.onlinelibrary.wiley.com/doi/abs/10.1029/2008JD010523>. <https://agupubs.onlinelibrary.wiley.com/doi/pdf/10.1029/2008JD010523>."

URL appeared twice.

11. Line 471-474

"(2020). URL <https://agupubs.onlinelibrary.wiley.com/doi/abs/10.1029/2020GL089688>.

<https://agupubs.onlinelibrary.wiley.com/doi/pdf/10.1029/2020GL089688>. [E2020GL089688](https://agupubs.onlinelibrary.wiley.com/doi/pdf/10.1029/2020GL089688)

<https://agupubs.onlinelibrary.wiley.com/doi/pdf/10.1029/2020GL089688>."

URL appeared twice.

12. Line 479-480

"2003). URL <https://agupubs.onlinelibrary.wiley.com/doi/abs/10.1029/2002JD002775>.

<https://agupubs.onlinelibrary.wiley.com/doi/pdf/10.1029/2002JD002775>.

<https://agupubs.onlinelibrary.wiley.com/doi/pdf/10.1029/2002JD002775>."

URL appeared twice.

13. Line 489-490

"(2020). URL <https://onlinelibrary.wiley.com/doi/abs/10.4905694/mja2.50545>.

<https://onlinelibrary.wiley.com/doi/pdf/10.5694/mja2.50545>."

URL appeared twice.

14 Line 492-493

URL appeared twice.

15. Reference section (overall):

Need to make a consistent references style. Journal names are sometimes written in short form and in some cases, full names are provided.

16. Figures (overall):

As a good practice, better to put the color bar's name. So I suggest putting the color bar's name.

17. Figure 3

"Wintertime climatology of tropospheric circulation and North African dust."

Is this a wintertime (extended winter months (NDJFM) in this case) climatology of tropospheric circulation or boreal winter climatological circulation? Please see my previous comment "6. Line 128"

Reviewer #2 (Remarks to the Author):

This paper proposes to present evidence of the impact of sudden stratospheric warmings (SSWs) on African dust export and the consequent impact on air quality, using chemical transport models' outputs and monitoring stations' observations.

The subject discussed in the present article is of high importance. The paper is fairly well structured but is quite long and hard to read. There are some points in the article that need to be clarified. The authors often generalize the results for the broad Mediterranean, which is not always possible nor correct (namely on section 2.3 which is based on data from 1 or 3 air quality monitoring stations), particularly as the transport for the Mediterranean is forced by different mechanisms and is affected also differently in the eastern and western regions. Some of the fundamental aspects of the methodology, namely the usage of different thresholds and limitation on monitoring station data, are very arguable and are not discussed in any way as a weakness of the approach. Some information on data is handled is missing, namely regarding monitoring stations. The number and placement of monitoring stations is also a caveat. Therefore, the methodology needs to be revised and missing information should be added. The results section on NAO (2.1) is disconnected from other results sections (2.2 and 2.3) and it is not mentioned on the discussion and abstract. Therefore, I would suggest revising the results and the connection to these sections. Moreover, the authors focus their attention on NAO disregarding any other teleconnection pattern without and clear justification (except that NAO as been shown to be related by the literature). At this point, I believe that this paper should NOT be considered for publication in its current form.

Below I point some comments and suggestions, which hopefully can help the authors to enhance the manuscript.

Comments:

1) Methods: Justify on why the seasonal cycle is defined as the first three Fourier harmonics of the daily climatology. This can be done differently. Please justify.

2) Methods: It would be important to see how the transport would affect stations located in the Southwestern are of Europe, like in the Iberian Peninsula or France. Records show that Iberia is often affected by SSS and with large impacts on mortality and the environment:

– Russo A., Sousa P.M., Durão R.M. Ramos A.M., Salvador P., Linares C., Diaz J., Trigo R.M. (2020) Saharan dust intrusions in the Iberian Peninsula: Predominant synoptic conditions. *Science of the Total Environment*, 717, 137041, doi: <http://doi.org/10.1016/j.scitotenv.2020.137041>

– Diaz J, Linares C, Carmona R, Russo A, Ortiz C, Salvador P, Trigo RM (2017) Saharan dust intrusions in Spain: Health impacts and associated synoptic conditions. *Environmental Research Volume 156*, July 2017, Pages 455-467, <https://doi.org/10.1016/j.envres.2017.03.047>

– J. Barkan, P. Alpert, H. Kutiel, P. Kishcha, Synoptics of dust transportation days from Africa toward Italy and central Europe, *J. Geophys. Res.*, 110 (7) (2005), pp. 1-14

– A.J. Fernández, M. Sicard, M.J. Costa, J.L. Guerrero-Rascado, J.L. Gómez-Amo, F. Molero, R. Barragán, S. Basart, D. Bortoli, A.E. Bedoya-Velásquez, M.P. Utrillas, P. Salvador, M.J. Granados-Muñoz, M. Potes, P. Ortiz-Amezcu, J.A. Martínez-Lozano, B. Artíñano, C. Muñoz-Porcar, R. Salgado, R. Román, F. Rocadenbosch, V. Salgueiro, J.A. Benavent-Oltra, A. Rodríguez-Gómez, L. Alados-Arboledas, A. Comerón, M. Pujadas, Extreme, wintertime Saharan dust intrusion in the Iberian Peninsula: Lidar monitoring and evaluation of dust forecast models during the February 2017 event, *Atmos. Res.*, 228 (2019), pp. 223-241, [10.1016/j.atmosres.2019.06.007](https://doi.org/10.1016/j.atmosres.2019.06.007)

– Sousa P., Barriopedro D., Ramos A.M., García-Herrera R., Espirito-Santo F., Trigo R.M. (2019) Saharan air intrusions as a relevant mechanism for Iberian heatwaves: The record breaking events of August 2018 and June 2019. *Weather and Climate Extremes*, 26, 100224, DOI: <http://doi.org/10.1016/j.wace.2019.100224>

Particularly, it would be important to test for these stations as they can be affected by atmospheric flows from different origins as shown by Bodenheimer et al. (2019) and Dayan et al (1991):

– S. Bodenheimer, I.M. Lensky, U. Dayan, Characterization of Eastern Mediterranean dust storms by area of origin; North Africa vs. Arabian Peninsula, *Atmos. Environ.* (2019), pp. 158-165, [10.1016/j.atmosenv.2018.10.034](https://doi.org/10.1016/j.atmosenv.2018.10.034)

– U. Dayan, J. Heffter, J. Miller, G. Gutman, Dust intrusion events into the Mediterranean Basin, *Journal of Applied Meteorology* (1988–2005), 30 (8) (1991), pp. 1185-1199

3) Methods: Does the air quality measure by the monitoring stations has missing values? How do you solve that problem? This is not mentioned anywhere

4) Methods (Lines 265-266): I didn't understand where did the 15 came from. From the 30-day periods? Please explain.

5) Methods/Results: Mortality changes exhibit a dipolar pattern with the centers moving towards large population centers. This is an expected outcome as the authors only account for the exposed population and don't differentiate between gender, age, comorbidities or even wealth conditions (e.g. access to air conditioners).

6) Methods: For Greece, the European Union air quality standard (daily mean PM₁₀ concentrations of 50µg m⁻³) is used. For Senegal, Mali and Niger, a relatively looser air quality standard (daily mean PM₁₀ concentrations of 250µg m⁻³) is used. I don't agree at all with the usage of different standards.

Although the number of exceeding days is higher in African countries is well known that the impacts in terms of human health are seen for lower values of PM concentrations. Moreover, it won't be comparable.

Please see:

[https://doi.org/10.1016/S2542-5196\(21\)00204-7](https://doi.org/10.1016/S2542-5196(21)00204-7)

doi: 10.1007/s40572-019-00235-7

<https://iopscience.iop.org/article/10.1088/1748-9326/abbf7a>

All of these highlight the fact that exposure below the air quality index breakpoints for good air Quality still have high impacts. So, the current standards for good air quality in the US and Europe (which are far more restrictive than the ones in Africa) need to be revised. Although that it may be legal to exceed the values in Africa, it does not mean that it does not affect people's health. Therefore, I see this as a major caveat and recommend on changing to the same threshold.

7) Methods: I think the section "Uncertainty in observed air quality responses" would fit better under Results than Methods.

8) Methods/Results (Ln60-61): Please revise this definition "In terms of the sea level pressure (SLP) field, a negative NAO can be defined as an anticyclonic anomaly centered around Stykkisholmur, Iceland and a cyclonic anomaly centered around Lisbon, Portugal35". For a more recent definition please see: <https://www.mdpi.com/2073-4433/11/5/544/htm>, <https://doi.org/10.5194/wcd-2020-20> , <https://doi.org/10.1002/wcc.150>

9) Methods/Results: Do other teleconnection patterns apart from NAO have influence on air quality and SSW influence on air quality? Please comment?

10) Results: The authors identified a cyclonic SLP anomaly over southern Europe-northern Africa, similar to the NAO-. As pointed in comment 7) the definition of negative phase of NAO should be revised. Therefore, the authors should revise the results following the abovementioned guidelines.

Moreover, the section on NAO is disconnected from other results and its not mentioned on the discussion and abstract. Therefore, I would suggest revising the results and the connection to these sections.

11) Lines 102-104 should be included in the discussion

12) The authors don't present a section highlighting the caveats of the study, and there are a few that should be mentioned (monitoring stations' data, thresholds, ...)

13) It is not clear to me, based on the analysis of figure 2a, that In the Mediterranean region, high dust concentrations tend to occur under the influence of southwesterly winds. Moreover, according to the literature, this not true for the all Mediterranean. At least, the authors should mention a contradictory example to the one cited focusing on Crete and discuss this.

Please see: <https://doi.org/10.1175/JAM2232.1>

Response to Reviewer #1

We would like to thank Reviewer 1 for the very helpful comments.

Please find below our responses to all of the reviewer's comments. The comments of Reviewer 1 are indicated in black and our responses are indicated in blue.

Reviewer #1 (Remarks to the Author):

-What are the noteworthy results?

The sudden stratospheric warming episodes set up a favorable environment at the tropospheric lower level over the North African region that enhances (suppresses) northward (southwestward) transport of dust in the Mediterranean region (West Africa). Subsequently, the transported dust leads to increased (decreased) premature death in the Mediterranean region (West Africa). The current work provides a way that can predict the impact of extreme stratospheric warming events on dust-related air pollution over southern Europe and West Africa very early (weeks to months before).

- Will the work be of significance to the field and related fields? How does it compare to the established literature? If the work is not original, please provide relevant references.

Yes. The current dust emission and transport modeling framework cannot accurately predict dust transport beyond a few days. So finding/understanding the way to predict dust transport out of the source region (here North Africa), as a result of sudden extreme stratospheric phenomena, well before (~weeks to months) is highly significant work in this field.

- Does the work support the conclusions and claims, or is additional evidence needed?

Yes, conclusions and claims are supported by the presented results/analyses.

- Are there any flaws in the data analysis, interpretation and conclusions? - Do these prohibit the publication or require revision?

I do not find any significant flaws in the data analysis that prohibit the publication. However, I have pointed out a few minor comments that need to be addressed before publication. These minor comments are provided below.

- Is the methodology sound? Does the work meet the expected standards in your field?

Yes.

- Is there enough detail provided in the methods for the work to be reproduced?

Yes.

Formal Review:

This article presented how sudden stratospheric warming (SSW) events enhance dust burden in the Mediterranean region and suppress dust burden in the West African region. Additionally, the article presented the importance of SSW signals for early prediction (~weeks to months before) of dust-related air pollution conditions in the Mediterranean and the West African regions. These forecasting strategies/signals play an important role as the current operational dust forecasting models cannot accurately forecast dust transport beyond a few days' limit. However, recent work has also been done in understanding the large-scale upper-level precursors (Rossby Wave Breaking signals) critical for organizing large-scale North African dust storms and subsequent long-range transport of dust to southern Europe. So, I encourage authors to look at these latest developments/findings that can provide an early signal of a high-impact dust outbreak.

I find this article very interesting and most of the interpretations are plausible. The paper is well written and the results and analyses are properly organized. However, there are a few minor issues that need to be addressed. Overall, I recommend this article for publication after addressing the minor comments shown below.

Minor comments

1. Line 23-24

“Apart from devastating health impacts, dust also impacts the environment, transport, and infrastructure”.

Please provide reference(s) to support the impact of dust on the environment, transport, and infrastructure.

Reply:

In the revised manuscript, we have added the following references (Goudie and Middleton 2006; World Bank 2019; Monteiro et al. 2022) to support the multi-sectoral impact of dust. Please see lines 27-28 of the revised manuscript, where we write:

“Apart from devastating health impacts, dust also impacts the environment, transport, and infrastructure¹²⁻¹⁴.”

REFERENCE

- Goudie, A. and Middleton, N. J.: Desert Dust in the Global System, Springer, Berlin, Heidelberg, <https://doi.org/10.1007/3-540-32355-4>, 2006.
- World Bank. 2019. Sand and Dust Storms in the Middle East and North Africa Region-Sources, Costs, and Solutions. Washington, DC.
- Monteiro et al. 2022. Multi-sectoral impact assessment of an extreme African dust episode in the Eastern Mediterranean in March 2018, *Science of The Total Environment*, Volume

843, 2022, 156861.

2. Line 30-31

“In many parts of Europe, today’s dust models are only capable of providing operational dust forecasts for several days in advance^{16,17}. Extending this to weeks”

Here it is worthy to note the latest developments and findings, the large-scale upper-level precursors (especially Rossby Wave Breaking signals), that can be a good indicator for early prediction of high-impact North African dust outbreaks over southern Europe.

Reply:

In the revised manuscript, we have added Orza et al. (2020) and Dhital et al. (2020) to the introduction section. Please see lines 36-41 of the revised manuscript, where we write:

“Case studies of North African dust storms with strong impact over southern Europe found a common large-scale upper-level precursor (a double Rossby wave breaking process) developing five to ten days prior to dust storm formation. Using this precursor signal as an indicator might provide early warnings of high-impact North African dust outbreaks over southern Europe at lead times of five to ten days^{20,21}. In general, the predictive lead time is limited to several days. Extending this lead time to weeks and even months would provide medical, agricultural, and nautical stakeholders more time to prepare for risks associated with airborne dust¹⁷.”

REFERENCE

Dhital S, ML Kaplan, JAG Orza, S Fiedler (2020). Atmospheric dynamics of a Saharan dust outbreak over Mindelo, Cape Verde Islands, preceded by Rossby wave breaking: Multiscale observational analyses and simulations. *Journal of Geophysical Research: Atmospheres*, 125, e2020JD032975.

Orza JAG, S Dhital, S Fiedler, ML Kaplan (2020). Large Scale Upper-level Precursors for Dust Storm Formation over North Africa and Poleward Transport to the Iberian Peninsula. Part I: An Observational Analysis. *Atmospheric Environment*, 237, 117688.

3. Line 34

“consider the impact of the stratosphere”

Is the impact of the stratosphere or the impact of the stratospheric phenomena. Please clearly state.

Reply:

In the revised manuscript, we have replaced “the stratosphere” with “stratospheric variability”.

Please see lines 42-43 of the revised manuscript, where we write:

“One potential approach towards filling this capability-need gap is to consider the impact of stratospheric variability,”

4. Line 51-52

“30% changes in aerosol concentrations caused by SSWs.”

Where aerosol concentration is changing? Please clarify this line.

Reply:

In the revised manuscript, we have added “over West Africa and the Eastern Mediterranean”. Please see lines 60-61 of the revised manuscript, where we write:

“30% changes in aerosol concentrations over West Africa and the Eastern Mediterranean”

5. Line 52-53

“SSWs induce those meteorological conditions that increase northward dust transport to the Mediterranean, while at the same time they inhibit African dust emission and decrease westward dust transport to West Africa.”

Is northward transport of dust solely African dust? I believe yes. If this is true, please use “northward African dust transport” just to make it consistent with your claim of inhibition of African dust emission later on.

Reply:

In the revised manuscript, we have made the corresponding changes. Please see lines 61-62 of the revised manuscript, where we write:

“SSWs induce meteorological conditions that increase northward African dust transport to the Eastern Mediterranean”

5. Line 87-90

“In the Mediterranean, SSWs lead to a 15-30% increase in the level of surface dust concentrations and a 20-40% increase in the number of high dust pollution days (Supplementary Table 1). In West Africa, decreases are simulated. The level of surface dust concentrations is reduced by 10-20%, and the number of high dust pollution days decreases between 10-30%, depending on the model and threshold used (Supplementary Table 1).”

As these % change values are exact without uncertainties (Supplementary Table S1). So I suggest using exact value ranges. For example, SSWs lead to an increase in dust concentration by 19-28% instead of 15-30%.

Reply:

In the revised manuscript, we have updated the Supplementary Table S1. In particular, for each region, we have added a column to show the combined range of the changes obtained from CESM2 and MERRA2. This combined range is to provide an estimate of the model uncertainty, which is obtained by taking the highest and lowest changes of the two models and then rounding these changes to the nearest ten. For each model, we now also show the sampling uncertainties in the SSW-caused changes as denoted by the subscript and superscript numbers. This involves randomly resampling, with replacement, 22 SSWs from the 22 available in CESM2 (or 24 SSWs from the 24 available in MERRA2), and recomputing the composite-mean SSW-caused changes 1000 times. The sampling uncertainty range is then taken as the 5th–95th percentile range. Please refer to the new Supplementary Table S1 for details.

Accordingly, we have made the corresponding changes to the manuscript. Please see lines 99-102 of the revised manuscript, where we write:

“In the Eastern Mediterranean, SSWs lead to a 20-30% increase in the level of surface dust concentrations and a 20-40% increase in the number of high dust pollution days (Supplementary Table 1). In West Africa, decreases are simulated. The level of surface dust concentrations is reduced by 10-20%, and the number of high dust pollution days decreases between 10-30%, depending on the model and threshold used (Supplementary Table 1).”

6. Line 128

“during boreal winter (Fig. 3a)”

Is this boreal winter or extended boreal winter months (NDJFM) that you have considered here? If not, please specify boreal winter months (which I believe comprise different months compared to “NDJFM”). In Figure 3a, you have mentioned: “(a) Wintertime climatology of tropospheric circulation and North African dust.” Is this wintertime consist “NDJFM” months or the boreal winter months? This need to be clearly mentioned because boreal winter months and extended boreal winter months are different here.

Reply:

In the revised manuscript, we have made the corresponding changes.

Please see line 148 of the revised manuscript, where we write:

“during the extended boreal winter season (NDJFM) (Fig. 3a),”

In the caption of Figure 3, we now write:

“(a) The extended wintertime (NDJFM) climatology of the tropospheric circulation and North African dust.”

7. Line 239

“daily mean zonal mean zonal wind”.

It should be “daily mean zonal wind”?

Reply:

In the definition of SSWs, the zonally-averaged zonal wind is used to detect the reversal of stratospheric westerly circumpolar winds. In lines 280 and 281 of the revised manuscript, we now write: *“daily-mean zonal-mean zonal wind”*.

8. Line 282

“While in general smaller particles (PM_{2.5}) are more important for health effects”.

Please provide reference(s) here.

Reply:

In the revised manuscript, we have changed the sentence. Please see lines 331-333 of the revised manuscript, where we write:

“While the smaller particles (PM_{2.5}) have been used in the previous section, because observations of these are not available we focus on observational records of PM₁₀ in this section as an indicator of air quality because long term observations of PM₁₀ are available for the studied areas.”

9. Line 399

“DUST STORMS URL”

Possibly full stop is missing here.

Reply:

In the revised manuscript, a full stop has been added.

10. Line 463-464

“search: Atmospheres 114 (2009). URL

<https://agupubs.onlinelibrary.wiley.com/doi/abs/10.1029/2008JD010523>.

<<https://agupubs.onlinelibrary.wiley.com/doi/abs/10.1029/2008JD010523>>

<https://agupubs.onlinelibrary.wiley.com/doi/pdf/10.1029/2008JD010523>.”

URL appeared twice.

Reply:

In the revised manuscript, the extra URL has been deleted.

11. Line 471-474

“(2020). URL <https://agupubs.onlinelibrary.wiley.com/doi/abs/10.1029/2020GL089688>.

E2020GL089688 2020GL089688, <https://agupubs.onlinelibrary.wiley.com/doi/abs/10.1029/2020GL089688>.

<<https://agupubs.onlinelibrary.wiley.com/doi/abs/10.1029/2020GL089688>>

[onlinelibrary.wiley.com/doi/pdf/10.1029/2020GL089688](https://agupubs.onlinelibrary.wiley.com/doi/pdf/10.1029/2020GL089688).”

URL appeared twice.

Reply:

In the revised manuscript, the extra URL has been deleted.

12. Line 479-480

“2003). URL <https://agupubs.onlinelibrary.wiley.com/doi/abs/10.1029/2002JD002775>.

<<https://agupubs.onlinelibrary.wiley.com/doi/abs/10.1029/2002JD002775>>

[com/doi/pdf/10.1029/2002JD002775.&](https://agupubs.onlinelibrary.wiley.com/doi/pdf/10.1029/2002JD002775)

<[https://agupubs.onlinelibrary.wiley.com/doi/pdf/10.1029/2002JD002775.&](https://agupubs.onlinelibrary.wiley.com/doi/pdf/10.1029/2002JD002775)>#x201D;

URL appeared twice.

Reply:

In the revised manuscript, the extra URL has been deleted.

13. Line 489-490

“(2020). URL <https://onlinelibrary.wiley.com/doi/abs/10.1029/2020GL089688>.

<<https://onlinelibrary.wiley.com/doi/abs/10.1029/2020GL089688>>

490 5694/mja2.50545.

<https://onlinelibrary.wiley.com/doi/pdf/10.5694/mja2.50545.&<https://onlinelibrary.wiley.com/doi/pdf/10.5694/mja2.50545.&>#x201D;>

URL appeared twice.

Reply:

In the revised manuscript, the extra URL has been deleted.

14 Line 492-493

URL appeared twice.

Reply:

In the revised manuscript, the extra URL has been deleted.

15. Reference section (overall):

Need to make a consistent references style. Journal names are sometimes written in short form and in some cases, full names are provided.

Reply:

In the revised manuscript, full names of the journals are now provided for all references.

16. Figures (overall):

As a good practice, better to put the color bar's name. So I suggest putting the color bar's name.

Reply:

In the revised manuscript, we have added the colorbar's name. Please see the updated Figure 1, Figure 2, Figure S2, and Figure S3 in the revised manuscript.

17. Figure 3

“Wintertime climatology of tropospheric circulation and North African dust.”

Is this a wintertime (extended winter months (NDJFM) in this case) climatology of

tropospheric circulation or boreal winter climatological circulation? Please see my previous comment “6. Line 128”

Reply:

The wintertime here refers to the extended winter season (NDJFM). We have made changes to the caption of Figure 3. Please see our reply to the above comment #6. In the revised manuscript, we have specified the winter months throughout the manuscript, wherever “wintertime” or “winter” is mentioned.

Response to Reviewer #2

We would like to thank Reviewer 2 for the very helpful comments.

Please find below our responses to all of the reviewer's comments. The comments of Reviewer 2 are indicated in black and our responses are indicated in blue.

Reviewer #2 (Remarks to the Author):

This paper proposes to present evidence of the impact of sudden stratospheric warmings (SSWs) on African dust export and the consequent impact on air quality, using chemical transport models' outputs and monitoring stations' observations.

The subject discussed in the present article is of high importance. The paper is fairly well structured but is quite long and hard to read. There are some points in the article that need to be clarified. The authors often generalize the results for the broad Mediterranean, which is not always possible nor correct (namely on section 2.3 which is based on data from 1 or 3 air quality monitoring stations), particularly as the transport for the Mediterranean is forced by different mechanisms and is affected also differently in the eastern and western regions. Some of the fundamental aspects of the methodology, namely the usage of different thresholds and limitation on monitoring station data, are very arguable and are not discussed in any way as a weakness of the approach. Some information on data is handled is missing, namely regarding monitoring stations. The number and placement of monitoring stations is also a caveat. Therefore, the methodology needs to be revised and missing information should be added. The results section on NAO (2.1) is disconnected from other results sections (2.2 and 2.3) and it is not mentioned on the discussion and abstract. Therefore, I would suggest revising the results and the connection to these sections. Moreover, the authors focus their attention on NAO disregarding any other teleconnection pattern without and clear justification (except that NAO as been shown to be related by the literature). At this point, I believe that this paper should NOT be considered for publication in its current form.

Below I point some comments and suggestions, which hopefully can help the authors to enhance the manuscript.

Comments:

1) Methods: Justify on why the seasonal cycle is defined as the first three Fourier harmonics of the daily climatology. This can be done differently. Please justify.

Reply:

In the below Figure R1 we show, for example, that at a location near Lisbon, Portugal, the annual cycle of the 35-year, daily climatology of zonal wind exhibits a substantial high-frequency component (black curve). This very likely reflects the fact that averaging over 35 years is not sufficient to reduce the sampling uncertainty associated with large amplitude

of weekly-timescale synoptic variability in the zonal wind at this location. This high-frequency variability can be removed by retaining only the mean and first three Fourier harmonics of the annual cycle (blue curve). This is a common approach used by many studies including Compo and Sardeshmukh (2004), Simpson et al. (2013), and Ogrosky and Stechmann (2015).

In the revised manuscript, we have added text to the “Methods” section to justify our use of the definition for the seasonal cycle. Please see lines 276-277 of the revised manuscript, where we write:

“The seasonal cycle is defined as the mean and first three Fourier harmonics of the daily climatology to remove the high-frequency sampling variability, following previous studies⁶¹⁻⁶³.”

Figure R1. The annual cycle of the 35-yr (1980-2014) daily climatology of the 10-meter zonal wind at a location near Lisbon, Portugal [351.25E,38.75N] (black curve). Also shown is the mean and first three Fourier harmonics of the annual cycle (blue curve). The 10-meter zonal wind field comes from the MERRA2 reanalysis.

REFERENCES

- Compo, G. P., & Sardeshmukh, P. D. (2004). Storm Track Predictability on Seasonal and Decadal Scales, *Journal of Climate*, **17**(19), 3701-3720.
- Ogrosky, H. R., & Stechmann, S. N. (2015). Assessing the Equatorial Long-Wave Approximation: Asymptotics and Observational Data Analysis, *Journal of the Atmospheric Sciences*, **72**(12), 4821-4843.
- Simpson, I. R., Hitchcock, P., Shepherd, T. G., & Scinocca, J. F. (2013). Southern Annular Mode Dynamics in Observations and Models. Part I: The Influence of Climatological Zonal Wind Biases in a Comprehensive GCM, *Journal of Climate*, **26**(11), 3953-3967.

2) Methods: It would be important to see how the transport would affect stations located in the Southwestern are of Europe, like in the Iberian Peninsula or France. Records show that Iberia is often affected by SSS and with large impacts on mortality and the environment:

– Russo A., Sousa P.M., Durão R.M. Ramos A.M., Salvador P., Linares C., Diaz J., Trigo R.M. (2020) Saharan dust intrusions in the Iberian Peninsula: Predominant synoptic conditions. *Science of the Total Environment*, 717, 137041, doi:

<http://doi.org/10.1016/j.scitotenv.2020.137041>

<<http://doi.org/10.1016/j.scitotenv.2020.137041>>

– Diaz J, Linares C, Carmona R, Russo A, Ortiz C, Salvador P, Trigo RM (2017) Saharan dust intrusions in Spain: Health impacts and associated synoptic conditions. *Environmental Research* Volume 156, July 2017, Pages 455-467,

<https://doi.org/10.1016/j.envres.2017.03.047><<https://doi.org/10.1016/j.envres.2017.03.047>>

– J. Barkan, P. Alpert, H. Kutiel, P. Kishcha, Synoptics of dust transportation days from Africa toward Italy and central Europe, *J. Geophys. Res.*, 110 (7) (2005), pp. 1-14

– A.J. Fernández, M. Sicard, M.J. Costa, J.L. Guerrero-Rascado, J.L. Gómez-Amo, F. Molero, R. Barragán, S. Basart, D. Bortoli, A.E. Bedoya-Velásquez, M.P. Utrillas, P. Salvador, M.J. Granados-Muñoz, M. Potes, P. Ortiz-Amezcu, J.A. Martínez-Lozano, B. Artíñano, C. Muñoz-Porcar, R. Salgado, R. Román, F. Rocadenbosch, V. Salgueiro, J.A. Benavent-Oltra, A. Rodríguez-Gómez, L. Alados-Arboledas, A. Comerón, M. Pujadas, Extreme, wintertime Saharan dust intrusion in the Iberian Peninsula: Lidar monitoring and evaluation of dust forecast models during the February 2017 event, *Atmos. Res.*, 228 (2019), pp. 223-241, 10.1016/j.atmosres.2019.06.007

– Sousa P., Barriopedro D., Ramos A.M., García-Herrera R., Espirito-Santo F., Trigo R.M. (2019) Saharan air intrusions as a relevant mechanism for Iberian heatwaves: The record breaking events of August 2018 and June 2019. *Weather and Climate Extremes*, 26, 100224,

DOI: <http://doi.org/10.1016/j.wace.2019.100224>

<<http://doi.org/10.1016/j.wace.2019.100224>>

Particularly, it would be important to test for these stations as they can be affected by atmospheric flows from different origins as shown by Bodenheimer et al. (2019) and Dayan et al (1991):

– S. Bodenheimer, I.M. Lensky, U. Dayan, Characterization of Eastern Mediterranean dust storms by area of origin; North Africa vs. Arabian Peninsula, *Atmos. Environ.* (2019), pp. 158-165, 10.1016/j.atmosenv.2018.10.034

– U. Dayan, J. Heffter, J. Miller, G. Gutman, Dust intrusion events into the Mediterranean Basin, *Journal of Applied Meteorology* (1988–2005), 30(8) (1991), pp. 1185-1199

Reply:

We would like to thank the reviewer for the very helpful references. We fully agree with the reviewer that Saharan dust intrusions could affect the Iberian Peninsula under certain

meteorological conditions. For example, Russo et al. (2020) found that Saharan dust intrusions into central and southern Iberia are more likely to occur under a south-easterly wind regime, which is responsible for the establishment of dust transport from the Saharan region towards Iberia.

Under the meteorological conditions associated with SSWs, there is no dust signal in the Iberian Peninsula, as can be seen in the below Fig. R2 (or Fig. 1c and Fig. S2a in the manuscript). This is because the surface circulation response to SSWs corresponds, on average, to a negative NAO pattern, of which the cyclonic center is associated with a south-westerly component regime (please see the below Fig. R3 or Fig. 1a in the manuscript). The south-westerly wind favors a dust transport from the Saharan region towards the Eastern Mediterranean, rather than into the Iberian Peninsula.

Another potential reason for the lack of enhanced dust signal over the Iberian Peninsula is the increased rainfall in this region following SSW events. As can be seen in the below Fig. R4 or Fig. 2b in Domeisen and Butler (2020) <<https://doi.org/10.1038/s43247-020-00060-z>>, SSWs are associated with increased rainfall over the Iberian Peninsula and the Western Mediterranean. The washout effect may prevent the increase of airborne desert dust load over the Iberian Peninsula and the Western Mediterranean.

To sum up, the impact of SSWs on surface dust is largely limited to the Eastern Mediterranean, rather than over the broad Mediterranean. Therefore, in the revised manuscript, we have replaced “*Mediterranean*” with “*Eastern Mediterranean*” to specify the region affected by SSWs.

We also fully agree with the reviewer that the Eastern Mediterranean is exposed to dust originating from two large sources: a western source, namely North Africa (mainly Sahara Desert) and an eastern source, the Arabian Peninsula. It has been shown in Bodenheimer et al. (2019) <<https://doi.org/10.1016/j.atmosenv.2018.10.034>> that Saharan dust outbreaks affecting the Eastern Mediterranean occur mostly during winter, while the eastern dust storms originating from the Arabian Peninsula occur mainly during fall. Given the fact that SSW events occur during winter, and the fact that their occurrence is followed by a significant increase in the fraction of southwesterly winds, it is more likely that the enhanced dust signal over the Eastern Mediterranean during SSW episodes arises from the increased northward Saharan dust transport. In the revised manuscript, we have added a discussion on the different dust origins. Please see lines 130-138 of the revised manuscript, where we write:

“In this sense, the enhanced dust burden over the Eastern Mediterranean during SSW episodes (Fig. 1c) is likely related to the increased northward Saharan dust transport induced by an increase in the fraction of southwesterly winds (Fig. 2b). In terms of the surface wind speeds, however, there are no significant changes due to the SSW (Supplementary Figure 4a). Note that the Eastern Mediterranean is also exposed to dust originating from western Asia (mainly the Arabian Peninsula). Therefore, Arabian Desert dust imported under an easterly flow can also lead to dust intrusions into the Eastern Mediterranean^{44–46}. However, during SSW episodes, a significant decrease in the fraction of easterly wind is seen (Fig. 2b),

indicating that the enhanced dust burden over the Eastern Mediterranean during SSW episodes is unlikely contributed by dust from the Arabian Desert. There is no evidence for changes in dust burden due to the SSW in the Western Mediterranean basin (Fig. 1c).”

Figure R2. The large-scale dust response to SSWs from (left) CESM2 and (right) MERRA2. The left and right panels are the same as Fig. 1c and Fig. S2a in the manuscript, respectively.

Figure R3. The large-scale atmospheric circulation response to SSWs. This panel is the same as Fig. 1a in the manuscript.

Figure R4. The surface rainfall response to SSWs. This panel is the same as Fig. 2b in Domeisen and Butler (2020) <<https://doi.org/10.1038/s43247-020-00060-z>>.

3) Methods: Does the air quality measure by the monitoring stations has missing values? How

do you solve that problem? This is not mentioned anywhere

Reply:

The air quality measurements by the monitoring stations do have missing values, probably due to electricity failures, breakdowns of the network, or instrument maintenance. These missing values have been discarded. This is unlikely to affect the representativity of the measurements given their high recovery rate. For example, Marticorena et al. (2010) <<https://doi.org/10.5194/acp-10-8899-2010>> have shown that the annual recovery rate for PM₁₀ concentrations ranges between 92 and 99% for the three stations in western Africa.

In the revised manuscript, we have added a description of how we deal with the missing values and the rationale behind it. Please see lines 266-268 of the revised manuscript, where we write:

“Unlike the model outputs, the observational data are not recorded every day due to network and/or power failures, instrument maintenance, etc. In this study, the days with missing values are discarded. This is unlikely to affect the representativity of the measurements given their high recovery rate⁴⁸.”

4) Methods (Lines 265-266): I didn’t understand where did the 15 came from. From the 30-day periods? Please explain.

Reply:

The 15 here indicates the number of 2-day periods over days 1-30 after the onset of SSWs. For example, the 1st 2-day period corresponds to days 1-2, the 2nd 2-day period corresponds to days 3-4, the 3rd 2-day period corresponds to days 5-6, and the 15th 2-day period corresponds to days 29-30. Please see the below Table R1.

Table R1. The 30 days after the onset of SSWs are divided into 15 two-day periods.

Days	1-2	3-4	5-6	7-8	9-10	11-12	13-14	15-16	27-28	29-30
Number of 2-day periods	1	2	3	4	5	6	7	8	14	15

In the revised manuscript, we have added a brief clarification on this. Please see lines 313-315 of the revised manuscript, where we write:

“The 15 here indicates the number of 2-day periods over days [1,30] after the onset of SSWs. For example, the 1st 2-day period corresponds to days [1,2], the 2nd 2-day period corresponds to days [3,4], the 3rd 2-day period corresponds to days [5,6], and the 15th 2-day period corresponds to days [29,30].”

5) Methods/Results: Mortality changes exhibit a dipolar pattern with the centers moving towards large population centers. This is an expected outcome as the authors only account for the exposed population and don't differentiate between gender, age, comorbidities or even wealth conditions (e.g. access to air conditioners).

Reply:

As noted by the reviewer, the mortality impacts are weighted by population, and thus the impact of the SSWs shifts from the highest concentration changes to the population centers with high aerosol changes. Subtleties having to do with gender, age, comorbidities are unlikely to modify this large scale feature, since there are still more people to be impacted in populated regions. However, air pollution impact epidemiology studies are not able to disentangle these subtleties, and thus we use the standard air pollution relationships for this study.

In the revised manuscript, we add in a sentence to lines 309-311, where we write:

“There are uncertainties in the β deduced as the concentration-response factor, as well as what baseline mortality to use in the exposure by gender, age or comorbidities, but here we use a standard relationship and standard mortality values used for air pollution impacts⁶⁶.”

6) Methods: For Greece, the European Union air quality standard (daily mean PM10 concentrations of 50 μgm^{-3}) is used. For Senegal, Mali and Niger, a relatively looser air quality standard (daily mean PM10 concentrations of 250 μgm^{-3}) is used. I don't agree at all with the usage of different standards.

Although the number of exceeding days is higher in African countries is well known that the impacts in terms of human health are seen for lower values of PM concentrations. Moreover, it won't be comparable.

Please see:

<https://doi.org/10.1016/S2542-5196>

<[https://doi.org/10.1016/S2542-5196\(21\)00204-7](https://doi.org/10.1016/S2542-5196(21)00204-7)>

doi: 10.1007/s40572-019-00235-7

<https://iopscience.iop.org/article/10.1088/1748-9326/abbf7a>

<<https://iopscience.iop.org/article/10.1088/1748-9326/abbf7a>>

All of these highlight the fact that exposure below the air quality index breakpoints for good air Quality still have high impacts. So, the current standards for good air quality in the US and Europe (which are far more restrictive than the ones in Africa) need to be revised. Although that it may be legal to exceed the values in Africa, it does not mean that it does not affect people's health. Therefore, I see this as a major caveat and recommend on changing to the same threshold.

Reply:

We would like to thank the reviewer for the very helpful references. To address this comment, first of all, we would like to highlight that while the changes in the number of poor air quality days depend on the air quality standard (AQS), the changes in the level of PM₁₀ concentrations due to the SSW are independent of AQS. As can be seen in Fig. 4b and Fig. 4f in the revised manuscript, SSWs lead to a significant increase in the level of PM₁₀ concentrations in Greece and a significant decrease in the level of PM₁₀ concentrations in Senegal. These impacts are robust because they are independent of the AQSs.

As to the changes in the number of poor air quality days, as suggested by the reviewer, in the revised manuscript, we have used the same threshold (both 50 $\mu\text{g m}^{-3}$ and 260 $\mu\text{g m}^{-3}$) for Greece and Senegal, and these do not have a substantial impact on our results. For Greece, using both standards, SSWs lead to a significant increase in the number of poor air quality days (Fig. 4c-d in the revised manuscript), consistent with the significant increase in the level of PM₁₀ concentrations (Fig. 4b). Figure 4d also shows that when using 260 $\mu\text{g m}^{-3}$ as the standard for Greece, the number of poor air quality days per 30-day period is always less than 1 day. This is because the level of PM₁₀ concentrations in Greece is too low to be raised up to 260 $\mu\text{g m}^{-3}$ (see the range of x-axis of Fig. 4b). For Senegal, when using 260 $\mu\text{g m}^{-3}$ as the standard, there is a significant decrease in the number of poor air quality days due to the SSW (Fig. 4h), consistent with the significant decrease in the level of PM₁₀ concentrations (Fig. 4f). By contrast, when using 50 $\mu\text{g m}^{-3}$ as the standard for Senegal, there are no significant changes in the number of poor air quality days due to the SSW (Fig. 4g). This may be because the level of PM₁₀ concentrations in Senegal is too high to be reduced to 50 $\mu\text{g m}^{-3}$ (see the range of x-axis of Fig. 4f). Figure 4g also shows that when using 50 $\mu\text{g m}^{-3}$ as the standard for Senegal, the number of poor air quality days per 30-day period can reach up to 30 days (see the non-SSW spread in Fig. 4g; indicated by green bars).

To further test the robustness of our results, for Greece and Senegal where SSWs exhibit significant impacts on the level of PM₁₀ concentrations, we have calculated the changes in the number of poor air quality days defined over a wide range of AQSs. The AQSs used range from 10 $\mu\text{g m}^{-3}$ to 300 $\mu\text{g m}^{-3}$ with an increment of 10 $\mu\text{g m}^{-3}$. The results are shown in the below Fig. R5 (and the Supplementary Figure 6 in the revised manuscript). The thick pink curve indicates the number of poor air quality days per 30-day period during SSW episodes. At a given threshold value, the shadings show the spread of the number of poor air quality days per 30-day period during 1000 sets of non-SSW episodes, with the thick black curve indicating the non-SSW mean and the dashed black curves indicating the 5th-95th confidence interval. First of all, for both the SSW composite (thick pink curve) and the non-SSW mean (thick black curve), the number of poor air quality days decreases as the threshold value increases. Meanwhile, the SSW composite (thick pink curve) and the non-SSW mean (thick black curve) are well separated from each other among most of the threshold values. In particular, for Greece, the SSW composite (thick pink curve) lies to the east of the non-SSW mean (thick black curve) (Fig. R5, upper panel), indicating that the number poor air quality days in Greece increases due to the SSW; for Senegal, the SSW composite (thick pink curve)

lies to the west of the non-SSW mean (thick black curve) (Fig. R5, lower panel), indicating that the number of poor air quality days in Senegal decreases due to the SSW. These results suggest that the impacts of SSWs on the number of poor air quality days are robust among a wide range of AQSs. The lower panel in Fig. R5 also shows that, in Senegal, at threshold values around $50 \mu\text{g m}^{-3}$ where the number of poor air quality days per 30-day period can reach up to 30 days (see the shadings in Fig. R5, lower panel), the SSW composite (thick pink curve) and the non-SSW mean (thick black curves) are not well separated from each other, indicating that SSWs are unlikely to lower the level of PM_{10} concentrations in Senegal to the EU standard $50 \mu\text{g m}^{-3}$. This is consistent with the results shown in Fig. 4g in the revised manuscript.

To sum up, SSWs lead to a significant increase in the level of PM_{10} concentrations in Greece and a significant decrease in the level of PM_{10} concentrations in Senegal. These impacts are robust because they are independent of the air quality standards. In terms of the changes in the number of poor air quality days due to the SSW, the increase in Greece and the decrease in Senegal are robust over a wide range of air quality standards. These findings suggest that our results are not sensitive to the air quality standards.

In the revised manuscript, we have added a discussion on this. Please see lines 172-192 of the revised manuscript, where we write:

“To examine whether this expectation holds, two air quality indicators are considered: (a) the level of PM_{10} concentrations, and (b) the expected number of poor air quality days when PM_{10} concentrations exceed the given thresholds (two thresholds are considered, see Methods). In Greece, all air quality indicators during SSW episodes (black vertical lines in Fig. 4b-d) lie entirely beyond the upper end of the non-SSW spread (color histograms in Fig. 4b-d), indicating that the air quality in Greece is worsened significantly during SSW episodes. In particular, the level of PM_{10} concentrations reaches $31 \mu\text{g m}^{-3}$ (black vertical line in Fig. 4b), an increase of 64% relative to the non-SSW mean ($19 \mu\text{g m}^{-3}$; purple vertical line in Fig. 4b); the expected number of poor air quality days per 30-day period exceeding $50 \mu\text{g m}^{-3}$ reaches 3.8 days (black vertical line in Fig. 4c), roughly four times the non-SSW mean (one day; green vertical line in Fig. 4c); the expected number of poor air quality days per 30-day period exceeding the higher threshold of $260 \mu\text{g m}^{-3}$ also increases by a factor of four (Fig. 4d), though PM_{10} concentrations at this station rarely reach this level. In Senegal, the level of PM_{10} concentrations and the expected number of poor air quality days exceeding the threshold of $260 \mu\text{g m}^{-3}$ decrease significantly during SSW episodes (Fig. 4f,h), indicating that the air quality in Senegal is improved considerably due to the SSW. In particular, the level of PM_{10} concentrations is $136 \mu\text{g m}^{-3}$ (black vertical line in Fig. 4f), a decrease of 34% relative to the non-SSW mean ($207 \mu\text{g m}^{-3}$; purple vertical lines in Fig. 4f); the expected number of poor air quality days exceeding the threshold of $260 \mu\text{g m}^{-3}$ decreases to 2.4 days (black vertical line in Fig. 4h), about 34% of the non-SSW mean (7.0 days; green vertical line in Fig. 4h). The lower threshold of $50 \mu\text{g m}^{-3}$ is nearly always exceeded at this station, and no significant changes are seen during SSW episodes in this case (Fig. 4g). We note nonetheless that changes in the expected number of poor air quality days during SSW episodes are robust across a wide range of thresholds (Supplementary Figure 6), and that exposure to air

pollutants at any level has observable health impacts⁴⁹⁻⁵¹.”

In the revised manuscript, we also make direct reference to the health impacts of permissible concentrations of air pollutants and the necessity of revising the current standards for good air quality. Please see lines 346-348 of the revised manuscript, where we write:

“Note that exposure to permissible concentrations of air pollutants still has observable health impacts⁴⁹⁻⁵¹, indicating that the air quality standards should be regularly reviewed and revised as new scientific evidence emerges on adverse effects on public health and the environment.”

Figure R5. The expected number of poor air quality days per 30-day period in (upper) Greece and (lower) Senegal. At a given threshold value, poor air quality days are defined when the daily mean PM₁₀ concentrations exceed this standard. The standards used range from 10 µg m⁻³ to 300 µg m⁻³ with an increment of 10 µg m⁻³ (see the y-axis). The thick pink curve indicates the expected number of poor air quality days per 30-day period during SSW episodes. For each threshold value, the shadings show the spread of the number of poor air quality days per 30-day period during 1000 sets of non-SSW episodes, with the thick black curve indicating the non-SSW mean and the dashed black curves indicating the 5th-95th percentile confidence interval derived from the non-SSW spread.

7) Methods: I think the section “Uncertainty in observed air quality responses” would fit better under Results than Methods.

Reply:

We thank the reviewer for this helpful suggestion. The reason for us to place it in the “Methods” section is that the main body of the manuscript is already quite long. In the revised manuscript, we have moved this section to the supplementary information. Please see

Supplementary Note 1 of the revised manuscript. We have also made the corresponding changes to the “Methods” section. Please see lines 268-270 of the revised manuscript, where we write:

“Note that the potential impacts of sampling error from short observational records and limited background stations may represent a potential source of uncertainty in observed air quality responses (Supplementary Note 1).”

8) Methods/Results (Ln60-61): Please revise this definition “In terms of the sea level pressure (SLP) field, a negative NAO can be defined as an anticyclonic anomaly centered around Stykkisholmur, Iceland and a cyclonic anomaly centered around Lisbon, Portugal³⁵”. For a more recent definition please see: <https://www.mdpi.com/2073-4433/11/5/544/htm> <<https://www.mdpi.com/2073-4433/11/5/544/htm>>, <https://doi.org/10.5194/wcd-2020-20> <<https://doi.org/10.5194/wcd-2020-20>> , <https://doi.org/10.1002/wcc.150> <<https://doi.org/10.1002/wcc.150>>

Reply:

We would like to apologize for the confusion here. The sentence pointed out by the reviewer is a description of the negative NAO pattern, not a definition of the negative NAO. In the revised manuscript, we have replaced “*can be defined as*” in this sentence with “*corresponds to*”. Please see lines 70-71 of the revised manuscript, where we write:

“In terms of the sea level pressure (SLP) field, a negative NAO corresponds to an anticyclonic anomaly at Stykkisholmur, Iceland and a cyclonic anomaly at Lisbon, Portugal³⁹.”

As to the definition of the NAO, it is defined in the caption of Figure 1, which is “*NAO index defined from the difference between normalized SLP between Lisbon, Portugal, and Stykkisholmur, Iceland*”. This is a very standard definition, and is exactly the same definition as used in one of the references suggested by the reviewer <<https://doi.org/10.5194/wcd-2020-20>>. This is also the definition used in Moulin et al. (1997) <<https://doi.org/10.1038/42679>> that demonstrated the role of the NAO in controlling the North African dust export. In the revised manuscript, we have moved the definition of the NAO index from the figure caption to the “Methods” section. Please see lines 285-288 of the revised manuscript, where we write:

“The impact of SSWs on the near-surface circulation is often described as resembling a negative phase of the North Atlantic Oscillation (NAO)³⁰⁻³³. In this study, the NAO index is defined from the difference between normalized SLP between Lisbon, Portugal, and Stykkisholmur, Iceland³⁹. This definition has been used in a previous study to demonstrate the role of the NAO in controlling the North African dust export².”

9) Methods/Results: Do other teleconnection patterns apart from NAO have influence on air

quality and SSW influence on air quality? Please comment?

Reply:

The reason for us to emphasize the impact of the NAO on African dust export is simply because this impact is closely related to the topic of this study. The impact of SSWs on the near-surface circulation is often described as resembling a negative NAO pattern, and the NAO has been associated with air quality impacts. However, we do not intend to imply that the NAO is the only teleconnection pattern that has an influence on air quality. For example, the El Niño-Southern Oscillation (ENSO) teleconnection also influences air quality in certain regions, such as wintertime PM_{2.5} pollution over China <<https://doi.org/10.1016/j.aosl.2022.100189>>, PM₁₀ concentrations on the Korean Peninsula <<https://doi.org/10.1016/j.atmosenv.2017.08.052>>, summertime surface ozone over the eastern United States <<https://doi.org/10.1002/2017GL076150>>, as well as Carbon Monoxide over the North Atlantic European Region in spring <<https://doi.org/10.3389/fenvs.2022.894779>>.

While the ENSO teleconnection has a broad influence on air quality over wide areas, the impacted pollutants and regions are beyond the interest of this study. By contrast, the NAO teleconnection is the one that has been shown to be closely linked to the SSW events and African dust export simultaneously, making it a unique pathway linking the extreme phenomenon in the stratosphere to the dust transport at the surface.

It is important to emphasize that in assessing the impact of SSWs on dust transport and air pollution, we have not made any assumptions about what teleconnection patterns are playing a role. We have stressed that the surface impact of SSWs is in fact stronger, more persistent, and somewhat larger in spatial extent than a projection on the NAO would imply. A more detailed investigation into the meteorology of why this is the case and whether it would be helpful to consider the role of other well-known teleconnection patterns would be interesting, but is beyond the scope of this study.

10) Results: The authors identified a cyclonic SLP anomaly over southern Europe-northern Africa, similar to the NAO-. As pointed in comment 7) the definition of negative phase of NAO should be revised. Therefore, the authors should revise the results following the abovementioned guidelines.

Moreover, the section on NAO is disconnected from other results and its not mentioned on the discussion and abstract. Therefore, I would suggest revising the results and the connection to these sections.

Reply:

In terms of the definition of the NAO, please refer to our reply to the above comment #8. As to the connection of the NAO section to other sections, we have made changes in the revised manuscript.

In the revised abstract (lines 5-7), we write:

“Here we show that on timescales of weeks to months, North African dust emission and transport are impacted by sudden stratospheric warmings (SSWs), which establish a negative North Atlantic Oscillation-like surface signal.”

In the revised discussion (lines 219-222), we write:

“The stratospheric impacts on dust transport and air pollution at the surface are driven by the role of SSWs in establishing a negative NAO-like signal at the surface. This NAO phase is typically associated with a weakened subtropical ridge, which, through increasing southwesterly winds in the Eastern Mediterranean and weakening northeasterly trade winds in West Africa, creates large-scale meteorological conditions favorable for the dipolar dust response.”

In the revised section “Meteorological causes of the dipolar dust response” (lines 153-154), we write:

“During SSW episodes, a cyclonic SLP anomaly is observed in the region of southern Europe / northern Africa, which is a typical pressure center of action of the negative NAO phase (Fig. 1a).”

11) Lines 102-104 should be included in the discussion

Reply:

In the revised manuscript, we have moved this sentence to the discussion section. Please see lines 214-216 of the revised manuscript, where we write:

“The changes from an SSW event in dust pollution mortality are comparable to the changes in cold weather mortality in the UK caused by an SSW, which is 620 additional deaths per event⁴².”

12) The authors don't present a section highlighting the caveats of the study, and there are a few that should be mentioned (monitoring stations' data, thresholds, ...)

Reply:

In the revised manuscript, we have added discussions about the caveats in the corresponding places. These are not merged into a single section given their appearance in very different locations.

In terms of the stations' data, in lines 266-268 of the revised manuscript, we write:

“Unlike the model outputs, the observational data are not recorded every day due to network and/or power failures, instrument maintenance, etc. In this study, the days with missing values

are discarded. This is unlikely to affect the representativity of the measurements given their high recovery rate⁴⁸.”

In terms of the thresholds, in lines 346-348 of the revised manuscript, we write:

“Note that exposure to permissible concentrations of air pollutants still has observable health impacts⁴⁹⁻⁵¹, indicating that the air quality standards should be regularly reviewed and revised as new scientific evidence emerges on adverse effects on public health and the environment.”

13) It is not clear to me, based on the analysis of figure 2a, that In the Mediterranean region, high dust concentrations tend to occur under the influence of southwesterly winds. Moreover, according to the literature, this not true for the all Mediterranean. At least, the authors should mention a contradictory example to the one cited focusing on Crete and discuss this.

Please see: <https://doi.org/10.1175/JAM2232.1>

<<https://doi.org/10.1175/JAM2232.1>>

Reply:

In the revised manuscript, we have revised the sentence. Please see lines 123-125 of the revised manuscript, where we write:

“In the Eastern Mediterranean, the southwesterly winds, which facilitate northward Sahara desert dust transport, create a potential for high dust concentrations (Fig. 2a).”

We would like to thank the reviewer for the very helpful reference by Dayan and Levy (2005). It has been shown in Dayan and Levy (2005), as well as Dayan et al. (1991) and Bodenheimer et al. (2019) that, the region of interest in this study, i.e., the Eastern Mediterranean, is exposed to dust originating from two large sources: a western source, namely North African (mainly Sahara Desert) and an eastern source, the Arabian Peninsula. Therefore, in addition to the southwesterly winds transporting Sahara Desert dust towards the Eastern Mediterranean, the easterly winds transporting Arabian Desert dust can also lead to high dust concentrations over the Eastern Mediterranean. In the revised manuscript, we have added a discussion about this. Please see lines 130-138 of the revised manuscript, where we write:

“In this sense, the enhanced dust burden over the Eastern Mediterranean during SSW episodes (Fig. 1c) is likely related to the increased northward Saharan dust transport induced by an increase in the fraction of southwesterly winds (Fig. 2b). In terms of the surface wind speeds, however, there are no significant changes due to the SSW (Supplementary Figure 4a). Note that the Eastern Mediterranean is also exposed to dust originating from western Asia (mainly the Arabian Peninsula). Therefore, Arabian Desert dust imported under an easterly flow can also lead to dust intrusions into the Eastern Mediterranean⁴⁴⁻⁴⁶. However, during SSW episodes, a significant decrease in the fraction of easterly wind is seen (Fig. 2b), indicating that the enhanced dust burden over the Eastern Mediterranean during SSW episodes is unlikely contributed by dust from the Arabian Desert. There is no evidence for

changes in dust burden due to the SSW in the Western Mediterranean basin (Fig. 1c).”

REFERENCE

- S. Bodenheimer, I.M. Lensky, U. Dayan, Characterization of Eastern Mediterranean dust storms by area of origin; North Africa vs. Arabian Peninsula, Atmospheric Environment (2019), pp. 158-165, 10.1016/j.atmosenv.2018.10.034
- U. Dayan, J. Heffter, J. Miller, G. Gutman, Dust intrusion events into the Mediterranean Basin, Journal of Applied Meteorology (1988-2005), 30(8) (1991), pp. 1185-1199

Reviewer #1 (Remarks to the Author):

Overview

This is my second time reviewing this manuscript. In my previous review, I raised minor comments before accepting this manuscript for possible publication. All the comments raised in my previous review are properly addressed. After looking at the response to my comments and the final version of the manuscript, I see great improvement in the manuscript. So I recommend this manuscript for possible publication.

Reviewer #2 (Remarks to the Author):

The authors made a truly amazing effort to respond to all my questions and requests. The answers were very detailed and clear. I approve this manuscript for publication and highlight its importance for the area.